# SCALABLE ENSEMBLE DIVERSIFICATION FOR OOD GENERALIZATION AND DETECTION

## ABSTRACT

Training a diverse ensemble of models has several practical applications such as providing candidates for model selection with better out-of-distribution (OOD) generalization, and enabling the detection of OOD samples via Bayesian principles. An existing approach to diverse ensemble training encourages the models to disagree on provided OOD samples. However, the approach is computationally expensive and it requires well-separated ID and OOD examples, such that it has only been demonstrated in small-scale settings.

**Method.** This work presents a Hardness-based Diversification Regularizer (HDR) applicable to large-scale settings (e.g. ImageNet) that does not require OOD samples. Instead, HDR identifies hard training samples on the fly and encourages the ensemble members to disagree on these. To improve scaling, we show how to avoid the expensive computations in existing methods of exhaustive pairwise disagreements across models.

**Results.** We evaluate the benefits of diversification with experiments on ImageNet. First, for OOD generalization, we observe large benefits from the diversification in multiple settings including output-space (classical) ensembles and weight-space ensembles (model soups). Second, for OOD detection, we turn the diversity of ensemble hypotheses into a novel uncertainty score estimator that surpasses a large number of OOD detection baselines.

## 1 INTRODUCTION

Training an ensemble of diverse models is useful in multiple applications. Diverse ensembles are used to enhance out-of-distribution (OOD) generalization, where strong spurious features learned from the in-domain (ID) training data hinder generalization (Lee et al., 2023; Pagliardini et al., 2023; Teney et al., 2022a;b). By learning multiple hypotheses, the ensemble is given a chance to learn more predictive features that may otherwise be overshadowed by prominent non-robust and spurious features (Chen et al., 2024; Yashima et al., 2022). In Bayesian machine learning, diversification of the posterior samples has been studied as a means to improve the precision and efficiency of sample uncertainty estimates (D'Angelo & Fortuin, 2021; Wilson & Izmailov, 2020).

A common strategy to train a diverse ensemble is to introduce a diversification objective while training the models in the ensemble in parallel (D'Angelo & Fortuin, 2021; Lee et al., 2023; Pagliardini et al., 2023; Ross et al., 2020; Scimeca et al., 2023). The main loss (e.g. cross-entropy for classification) encourages the models to fit the labeled training, while the diversification loss encourages the models to disagree with one another on unlabelled OOD samples (Lee et al., 2023; Pagliardini et al., 2023) (Figure 1). The models are thus driven to discover different hypotheses that all explain the in-domain (ID) data but behave different out of distribution.

The above approaches to diversification rely on the availability of two distinct sets of data: labeled in-domain (ID) examples for the main training objective and unlabeled OOD examples for diversification.

The existing methods are moreover computationally expensive, and have thus only been tested on small-scale artificial settings where the data can be clearly delineated into ID and OOD sets (Lee et al., 2023; Pagliardini et al., 2023). To relax the latter requirement for separate ID and OOD sets, some attempts were made to generate OOD data for diversification synthetically (Scimeca et al., 2023). It is however still unclear how to apply these methods to realistic large-scale applications (e.g. ImageNet scale) where distinct OOD samples are not readily available.

This paper presents a Hardness-based Diversification Regularizer (HDR, Figure 1) that addresses the limitations of the existing approaches. We introduce three technical innovations. (1) Our method dynamically identifies hard samples from the training data on which the models are encouraged to disagree. (2) At each iteration, the diversification objective is applied only on a random pair of models, alleviating the computational cost of the exhaustive pairing from prior work. (3) The diversification objective is applied to deep networks by only affecting a small subset of last layers, further reducing the computational costs. Altogether, these innovations enable scaling up to realistic applications that were so far out of reach for the mentioned family of methods.

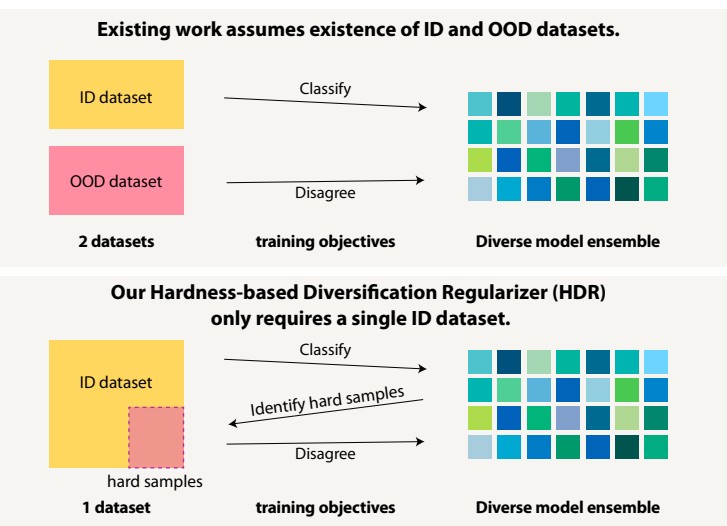

Figure 1: **Existing diversification methods (top)** require distinct (unlabeled) OOD examples on which the models are encouraged to disagree. Our **Hardness-based Diversification Regularizer (HDR, bottom)** instead encourages the models to diverge on hard examples identified within a single standard training set.

Our experiments evaluate HDR by training a diverse ensemble on ImageNet. We examine the benefits of the diversification for OOD generalization and OOD detection. For OOD generalization, we showcase the usage of HDR-diversified ensemble in three variants: (a) a classical ensemble (average of prediction probabilities) (Lakshminarayanan et al., 2017), (b) a model soup (average of model weights) (Wortsman et al., 2022), and (c) an oracle selection of the best individual model within the ensemble for each OOD test set (Lee et al., 2023; Teney et al., 2022a). In all three cases, HDR achieves superior generalization on multiple OOD test sets (ImageNet-A/R/C) (Hendrycks et al., 2021b;a; Hendrycks & Dietterich, 2019). For OOD detection, we examine multiple ways to use the HDR-diversified ensemble: (a) treating them as samples of the Bayesian posterior and (b) using a novel OODness estimate of Predictive Diversity Score (PDS) that measures the diversity of predictions from an ensemble. We show that PDS provides a superior detection of OOD samples like ImageNet-C, OpenImages (Wang et al., 2022), and iNaturalist (Huang & Li, 2021).

Our contributions are summarized as follows.

1. A novel Hardness-based Diversification Regularizer (HDR) that enables scaling up popular approaches to ensemble diversification based on prediction disagreement.

2. A novel Predictive Diversity Score (PDS) that estimates sample-wise OODness based on ensemble prediction diversity.

3. An empirical demonstration of ensemble diversification at the ImageNet scale, with demonstrated benefits in OOD generalization and detection.

## 2 PROPOSED METHOD

**Setting.** We denote our training data $\mathcal{D} := \{x_n, y_n\}_{n=1}^N$ and refer to it as the in-domain (ID) data. Prior diversification methods based on "prediction disagreement" Lee et al. (2023); Pagliardini et al. (2023) require a separate set of unlabeled out-of-distribution (OOD) examples $\mathcal{D}^{\text{ood}} := \{x_n^{\text{ood}}\}_{n=1}^{N^{\text{ood}}}$. Our proposed method will show how to proceed without $\mathcal{D}^{\text{ood}}$. We denote with $f(\cdot, \theta)$ a neural network classifier of parameters $\theta$. Then $f(x; \theta) \in \mathbb{R}^C$ corresponds to the logits over $C$ classes for the input $x$, and $p(x) := \text{Softmax}(f(x)) = \dfrac{e^{f(x)}}{\sum_{c=1}^C e^{f_c(x)}} \in [0,1]^C$ probabilities over the classes. Our goal is to obtain an ensemble $\{f^1, \cdots, f^M\}$ of $M$ models. Our experiments in Section 3 will showcase multiple ways to exploit these models (output-space ensembles, weight-space ensembles, etc.).

## 2.1 DIVERSIFICATION THROUGH DISAGREEMENT

The goal of our method is to train an ensemble of models that produce diverse predictions on $\mathcal{D}^{\text{ood}}$. Several methods were recently proposed to promote diversity in an ensemble by encouraging disagreement in the members' predictions by an auxiliary training objective (Lee et al., 2023; Pagliardini et al., 2023). These methods proceed by training a set of models $\{f^m\}_{m=1}^{m=M}$ in parallel. The main training objective is typically the cross-entropy loss over all $M$ ensemble members and $N$ training examples:

$$\mathcal{L}_{\text{main}} \;=\; \frac{1}{MN} \sum_n^N \sum_m^M -\log p_{y_n}^m(x_n; \theta). \tag{1}$$

While $\mathcal{L}_{\text{main}}$ encourages each ensemble to similarly fit the training data, an auxiliary disagreement objective is applied to every pair of models in the ensemble and every OOD example from $\mathcal{D}^{\text{ood}}$:

$$\mathcal{L}_{\text{div}} \;=\; \frac{1}{N^{\text{ood}} M(M-1)} \sum_{n=1}^{N^{\text{ood}}} \sum_{m=1}^{M} \sum_{l=1}^{m-1} \mathcal{G}\big(p^m(x_n^{\text{ood}}), \, p^l(x_n^{\text{ood}})\big). \tag{2}$$

The $\mathcal{G}(\cdot, \cdot)$ leads to diversification by encouraging a pair of models $(f^m, f^l)$ to disagree, i.e. make different predictions on samples from $\mathcal{D}^{\text{ood}}$. Our method applies to several implementations of $\mathcal{G}$ from the existing literature (D'Angelo & Fortuin, 2021; Lee et al., 2023; Pagliardini et al., 2023). In our experiments, $\mathcal{G}$ implements the A2D ("Agree to disagree") objective from (Pagliardini et al., 2023):

$$\mathcal{G}\big(p^m(x), \, p^l(x)\big) \;=\; -\log\Big( p_{\hat{y}}^m(x) \cdot \big(1 - p_{\hat{y}}^l(x)\big) \;+\; p_{\hat{y}}^l(x) \cdot \big(1 - p_{\hat{y}}^m(x)\big) \Big) \tag{3}$$

where $\hat{y} := \arg\max_c p_c^m(x)$ is the class predicted by the model $p^m$ (the definition could just as well use the prediction from $p^l$, which would make no practical difference (Pagliardini et al., 2023)). Minimizing (3) encourages $p^l$ to assign a lower likelihood to the class predicted by $p^m$ and vice versa.

The next sections present our Hardness-based Diversification Regularizer (HDR). It makes the concept of diversification through disagreement practically relevant, by eliminating the need for OOD data ($\mathcal{D}^{\text{ood}}$) and improving the computational cost. The technical innovations concern the dynamic selection of hard samples from the training data itself (§2.2) and the application of the disagreement objective to stochastic pairs of models as well as a limited model depth (§2.3).

## 2.2 DYNAMIC SELECTION OF HARD EXAMPLES

With no OOD data, it is difficult to apply disagreement methods since the main training objective encourages all models to fit the training examples, hence to *agree* on all available data. In practice, extra OOD for disagreement, which should clearly differ from the ID data, may not be readily available. Considering e.g. ImageNet as the training data, it is not even clear how to define and obtain data that qualifies as OOD or where the feature-label correlations clearly differ.

We sidestep these limitations by arguing that the reason OOD data are needed for diversification is not because they are out of distribution (i.e. out of the training dataset) but because they contain some hard data points that are needed to make the models diversify in plausible ways. Previous approaches sourced these data points from a separate dataset. We argue that such a dataset is not needed because eventually what we need are "hard" data points where models can plausibly differ in their responses.

For this reason, we propose to replace the OOD disagreement data with a set of hard training examples identified dynamically during training. The models are then encouraged to disagree on these examples. The desiderata for these hard samples are twofold: (a) we wish to discriminate samples where the ensemble members make mistakes and (b) we only trust the ensemble prediction for the hard sample identification when the ensemble is sufficiently trained.

We assign a sample-wise weight $\alpha_n$ to each training sample $(x_n, y_n) \in \mathcal{D}$:

$$\alpha_n := \frac{\text{CE}(\frac{1}{M} \sum_m f^m(x_n), y_n)}{\left( \frac{1}{|B|} \sum_{b \in B} \text{CE}(\frac{1}{M} \sum_m f^m(x_b), y_b) \right)^2} \tag{4}$$

where $\text{CE}(\frac{1}{M} \sum_m f^m(x_n), y_n)$ is the cross-entropy loss on the logit-averaged prediction and $B$ is a mini-batch that contains the sample $(x_n, y_n)$. $\alpha_n$ is a weight proportional to the ensemble loss on the sample, which fulfills

desideratum (a) mentioned above. The normalization then handles desideratum (b). To see this, consider the batch-wise weight:

$$\alpha_B := \frac{1}{|B|} \sum_{b \in B} \alpha_b = \frac{1}{\frac{1}{|B|} \sum_b \text{CE}(\frac{1}{M} \sum_m f^m(x_b), y_b)}. \tag{5}$$

Now $\alpha_B$ is *inversely proportional* to the average cross-entropy loss of the ensemble on the mini-batch $B$. Thus, the overall level of $\alpha_n$ for $n \in B$ is lower for earlier iterations of the ensemble training, where the predictions from the models are not trustworthy yet.

We now use the sample-wise weights $\alpha_n$ to define the HDR training objective:

$$\mathcal{L}_{\text{HDR}} := \mathcal{L}_{\text{main}} + \frac{\lambda}{NM(M-1)} \sum_n \sum_{m<l} \text{stopgrad}(\alpha_n) \cdot \mathcal{G}\big(p^m(x_n), p^l(x_n)\big), \tag{6}$$

where $\lambda > 0$ controls the strength of the diversification. The operator $\text{stopgrad}(\cdot)$ outputs a copy of its argument that is treated as a constant during backpropagation. Compared to Equation 2, this formulation does not require OOD disagreement data. Instead, all training examples are treated as potential hard samples to disagree on, and their difficulty is softly determined via $\alpha_n$.

**Justification for the adaptive weights.** To justify the adaptive nature of $\alpha_n$, let us examine the gradient of the total loss (Equation 6). Considering an ensemble of two models $m$ and $l$, we have the gradient of the loss on a sample $(x, y)$ w.r.t. the model $m$'s predicted probability for the ground truth class ($p_y^m(x)$) (see Appendix A.8 for details):

$$\nabla_{p_y^m(x)} \mathcal{L}_{\text{HDR}}(x, y) = -\frac{1}{p_y^m(x)} + \frac{\alpha_n(2p_y^l(x) - 1)}{C(m, l, y, x)}, \tag{7}$$

where the denominator $C(m, l, y, x)$ is some non-negative function that is upper-bounded by 1. The gradient consists of the two terms. The sign of the first one, which corresponds to the cross-entropy, is always negative. The sign of the second one, which corresponds to the disagreement objective, depends on the value of $p_y^l(x)$. The fact that the term can have different signs can cause training instabilities if none of the terms will dominate (have much higher absolute value comparing to another term) the total gradient.

The only way to control for that and avoid such instabilities is to make $\alpha_n$ proportional to $p_y^m(x)^{-1}$: $\alpha_n = \gamma p_y^m(x)^{-1}$ for some $\gamma > 0$. In such case the dominance of the total gradient by the second term is ensured when:

$$\frac{|2p_y^l(x) - 1|}{C(m, l, y, x)} \geq |2p_y^l(x) - 1| \geq \gamma^{-1}. \tag{8}$$

As a result, the total gradient will be lower on the correctly predicted samples and higher on the samples on which an ensemble makes mistakes (i.e. hard samples) while being dominated by the disagreement term for sufficiently high values of $\gamma$. This allows for ensemble diversification without harming the approximation of the training distribution. In practice, we make $\alpha_n$ proportional to $-\log p_y^m(x)$ in equation 4, as it exhibits better diversification than using $p_y^m(x)^{-1}$.

## 2.3 STOCHASTIC SUM AND SHALLOW DISAGREEMENT

Many diversification algorithms are based on exhaustive pairwise comparisons between all the models in the ensemble (see the second term of Equation 6). This scales quadratically with the size $M$ of the ensemble and makes ensemble training inefficient for higher values of $M$.

**Stochastic sum.** To overcome this quadratic scaling law we propose to use a stochastic sum. For every mini-batch $B$, we use a random subset of models $|\mathcal{I}| \in \{1, \cdots, M\}$ on which to compute the diversification term in Equation 6. In our experiments, we randomly sample one pair of models per batch ($|\mathcal{I}| = 2$). Such stochastic sum size allows to reduce training of an ensemble of 50 models from theoretical 663 GPU hours per epoch to 30 minutes per epoch (see Appendix A.7). In addition to the computational benefits, stochastic sums contribute to the ensemble diversity by exposing each member to different subsets of training data.

**Shallow disagreement.** To further speed up the training, we consider updating only a subset of the layers of the model with the HDR objective, keeping others frozen. More specifically, each ensemble member in the experiments of Section 3 is based on a frozen Deit3b model (Touvron et al., 2022) of which we diversify only the last two layers. Diversifying only the last layer results in worse performance presumably due to the convexity of the optimization problem (see Appendix A.1).

### 2.4 PREDICTIVE DIVERSITY SCORE (PDS) FOR OOD DETECTION

We now describe how to use diverse ensembles for OOD detection (Helton et al., 2004). This is based on evaluating the epistemic uncertainty, which is the consequence of the lack of training data in a given regions of the input space (Mukhoti et al., 2023; Hüllermeier & Waegeman, 2021). In these OOD regions, the lack of supervision means that diverse models are likely to disagree in their predictions (Malinin et al., 2019; Lee et al., 2023; Pagliardini et al., 2023). We therefore propose to use the *agreement rate* across models on a given sample to estimate the epistemic uncertainty and its "OODness".

**BMA Baseline.** Given an ensemble of models, a simple baseline for OOD detection is to compute the predictive uncertainty of the Bayesian Model Averaging (BMA) by treating the ensemble members as samples of the posterior $p(\theta|\mathcal{D})$ (Lakshminarayanan et al., 2017; Wilson & Izmailov, 2020):

$$\eta_{\text{BMA}} := \max_c \frac{1}{M} \sum_m p_c^m(x). \tag{9}$$

While being a strong baseline (Mukhoti et al., 2023) for OOD detection this notion of uncertainty does not directly exploit the potential diversity in individual models of the ensemble because it averages out the predictions along the model index $m$. In addition to that, mimicking the true distribution makes individual members have small values of $\max_c p_c^m(x)$ on training samples with high aleatoric uncertainty (Hüllermeier & Waegeman, 2021). This is why BMA is not a reliable indicator of epistemic uncertainty.

**Proposed Predictive Diversity Score (PDS).** We propose a novel measure for epistemic uncertainty that directly measures the prediction diversity of the individual members. Concretely,

$$\eta_{\text{PDS}} := \frac{1}{C} \sum_c \max_m p_c^m(x). \tag{10}$$

PDS is a continuous relaxation of the number of unique argmax predictions within an ensemble of models (#unique). To see this, consider the special case where $p^m \in \{0, 1\}$ are one-hot vectors. Then, $\max_m p_c^m(x)$ is 1 if any of $m$ predicts $c$ and 0 otherwise. Thus, in this example $\sum_c \max_m p_c^m(x)$ computes the number of classes predicted by at least one ensemble member. An illustrative case when PDS is preferable to BMA for epistemic uncertainty estimation can be seen in Figure 2.

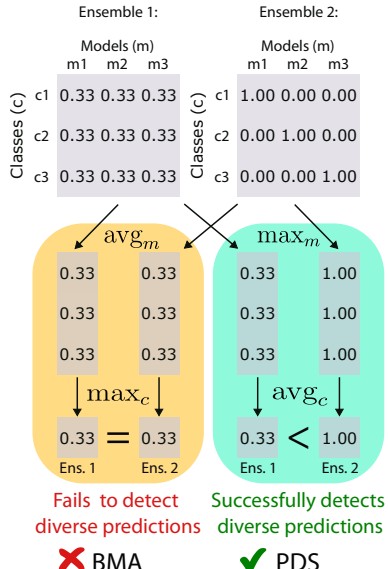

Figure 2: **BMA vs PDS**. Given sample $x$ and an ensemble of $M = 3$ models for $C = 3$ classes, which uncertainty scoring captures the ensemble diversity?

## 3 EXPERIMENTS

We present experiments that first evaluate the intrinsic diversification from HDR (§3.2) then evaluate several use cases of diverse ensembles for OOD generalization (§3.3) and OOD detection (§3.4).

### 3.1 EXPERIMENTAL SETUP

**Implementation.** For both tasks, we train an ensemble of models with HDR using the AdamW optimizer (Loshchilov & Hutter, 2019), a batch size varies from 16 to 256, learning rate from $10^{-4}$ to $10^{-3}$, weight decay is fixed to 0.01, and number of epochs to 10. The diversity weight $\lambda$ varies from 0 to 5 and the stochastic pairing is done for $|\mathcal{I}| = 2$ models for each mini-batch. All experiments use models based on the Deit3b architecture (Touvron et al., 2022) pretrained on ImageNet21k (Deng et al., 2009). As suggested in §2.3 we train only the last 2 layers. As in-domain (ID) data we use

the training split of ImageNet ( $|\mathcal{D}| = 1,281,167$). All experiments were run on RTX2080Ti GPUs with 12GB vRAM and 40GB RAM. Each experiment took between 2 to 12 hours.

**Baselines.** As a simple ensemble we use a variant of *deep ensembles* (Lakshminarayanan et al., 2017), which uses models trained independently with different random seeds.

To match the resource usage of our HDR, we also train only the last 2 layers of the models (i.e. they are "shallow ensembles").

We also consider simple ensembles of models with diverse hyperparameters (Wenzel et al., 2020). We reimplemented A2D (Pagliardini et al., 2023) and DivDis (Lee et al., 2023), with which we use unlabeled samples from ImageNet-R as disagreement data (the choice of dataset used for disagreement has little influence on the results, as seen in Table 9). For A2D, we use a frozen feature extractor and parallel training, i.e. all models are trained simultaneously rather than sequentially.

**Evaluation of OOD generalization.** We evaluate the classification accuracy of the ensembles trained on ImageNet with the (ID) validation split of ImageNet (IN-Val, 50,000 samples) and multiple OOD datasets: ImageNet-A (*IN-A* (Hendrycks et al., 2021b), 7.5k images & 200 classes), ImageNet-R (*IN-R* (Hendrycks et al., 2021a), 30k images, 200 classes), ImageNet-C (*IN-C-i* or just *C-i* for corruption strength $i$ (Hendrycks & Dietterich, 2019), 50k images, 1k classes). OpenImages-O (*OI* (Wang et al., 2022), 17k images, unlabeled), and iNaturalist (*iNat* (Huang & Li, 2021), 10k images, unlabeled).

**Evaluation of OOD detection.** The task is to differentiate samples from the above OOD datasets against those from the ImageNet validation data (considered as ID). The evaluation includes both "semantic" and "covariate" types of shifts (Zhang et al., 2023; Hendrycks & Dietterich, 2019; Hendrycks et al., 2021a; Recht et al., 2019; Yang et al., 2024). Openimages-O and iNaturalist represent semantic shifts because their label sets are disjoint from ImageNet's. And ImageNet-C represents a covariate shift because its label set is the same as ImageNet's but the style of images differs. We measure the OOD detection performance with the area under the ROC curve, following (Hendrycks & Gimpel, 2017).

## 3.2 DIVERSIFICATION

We start with the question of whether HDR truly diversifies the ensemble. To measure the diversity of the ensemble, we compute the number of unique predictions for each sample for the committee of models (#unique).

Table 1 shows the #unique and PDS values for the IN-Val as well as multiple OOD datasets. We observe that the deep ensemble baseline does not increase the diversity dramatically (e.g. 1.09 for IN-C-1) beyond no-diversity values (1.0). Diversification tricks like hyperparameter diversification (1.11 for IN-C-1) or A2D (1.04 for IN-C-1) only marginally change the prediction diversity. On the other hand, our HDR increases the prediction diversity across the board (e.g. 4.68 for iNat).

Qualitative results on ImageNet-R further verify the ability of HDR to diversify the ensemble (Figure 3). As a measure

| Method | IN-Val | IN-C-1 | IN-C-5 | iNat | OI |
|---|---|---|---|---|---|
| DE | 1.05 | 1.09 | 1.19 | 1.31 | 1.23 |
| +Div. HPs | 1.04 | 1.11 | 1.32 | 1.48 | 1.33 |
| A2D | 1.11 | 1.04 | 1.15 | 1.19 | 1.91 |
| HDR | **1.36** | **1.82** | **3.53** | **4.68** | **4.11** |

Table 1: **Diversity measure for ensembles.** We report the average #unique (number of unique classes among predictions of ensemble members for a given sample) on OOD datasets and IN-Val dataset (See §3.1 for the datasets). The ensemble size $M$ is 5 for all methods; $M$ is also the max possible #unique value. "+Div. HPs" stands for deep ensemble diversified via varying hyperparameters during its members' training.

for diversity, we use the Predictive Diversity Score (PDS) in §2.4. We observe that the samples inducing the highest diversity (high PDS scores) are indeed ambiguous: for the first image, where the "cowboy hat" is the ground truth category, we observe that "comic book" is also a valid label for the image style. On the other hand, samples with low PDS exhibit clearer image-to-category relationship.

## 3.3 OOD GENERALIZATION

We examine the first application of diverse ensembles: OOD generalization. We hypothesize that the superior diversification ability verified in §3.2 leads to greater OOD generalization due to the consideration of more robust hypotheses that do not rely on obvious spurious correlations.

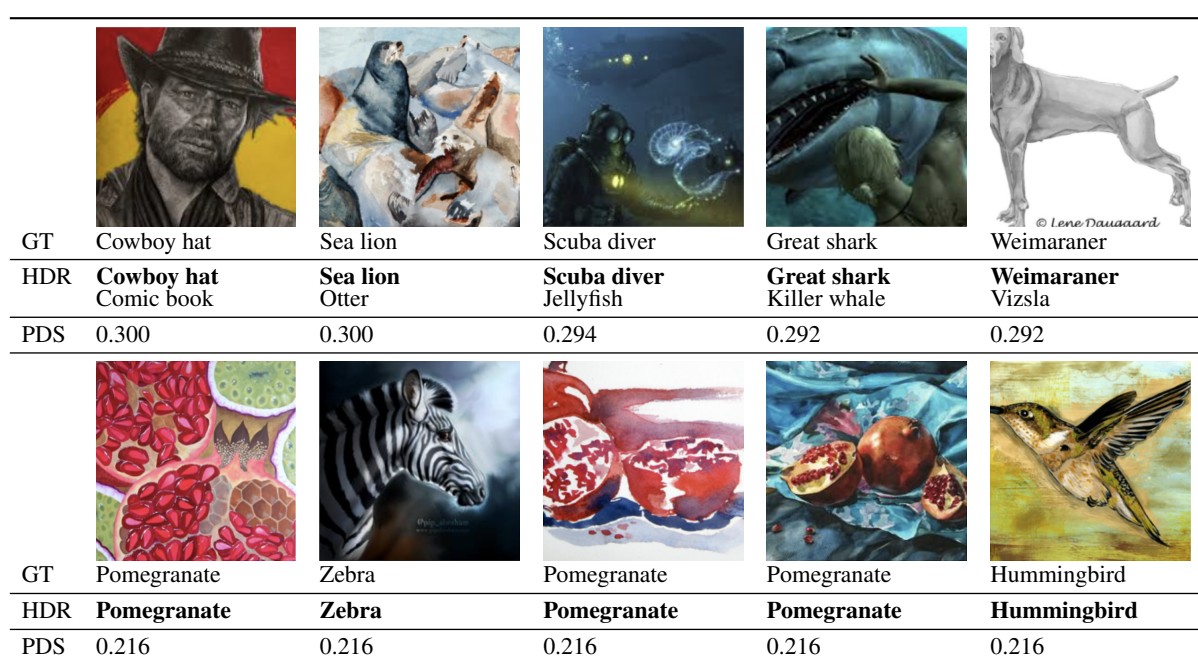

| | | | | |
|---|---|---|---|---|
| GT | Cowboy hat | Sea lion | Scuba diver | Great shark | Weimaraner |
| HDR | **Cowboy hat** Comic book | **Sea lion** Otter | **Scuba diver** Jellyfish | **Great shark** Killer whale | **Weimaraner** Vizsla |
| PDS | 0.300 | 0.300 | 0.294 | 0.292 | 0.292 |
| GT | Pomegranate | Zebra | Pomegranate | Pomegranate | Hummingbird |
| HDR | **Pomegranate** | **Zebra** | **Pomegranate** | **Pomegranate** | **Hummingbird** |
| PDS | 0.216 | 0.216 | 0.216 | 0.216 | 0.216 |

Figure 3: **ImageNet-R examples leading to the greatest and least disagreement**. We show the 5 most divergent and 5 least divergent samples according to the HDR ensemble. We measure prediction diversity with the Prediction Diversity Score (PDS) in §2.4. GT refers to the ground truth category. Ensemble predictions are shown in bold; in cases where ensemble members predict classes different from the ensemble prediction we provide them on the next line with standard font.

| Method | $M$ | Arch. | $\mathcal{D}_{\text{div}}$ | Prediction ensemble | | | | | Uniform Soup | | | | | Oracle Selection | | | | |
|---|---|---|---|---|---|---|---|---|---|---|---|---|---|---|---|---|---|---|
| | | | | Val | IN-A | IN-R | C-1 | C-5 | Val | IN-A | IN-R | C-1 | C-5 | Val | IN-A | IN-R | C-1 | C-5 |
| 1 model | 1 | Deit3b | - | 85.4 | 37.9 | 44.7 | 75.6 | 38.5 | 85.4 | 37.9 | 44.7 | 75.6 | 38.5 | 85.4 | 37.9 | 44.7 | 75.6 | 38.5 |
| DE | 5 | Deit3b | - | **85.4** | 39.9 | 46.3 | 75.7 | 38.6 | **85.3** | 36.7 | 44.6 | 75.5 | 38.3 | **85.4** | 37.9 | 44.9 | 75.7 | 38.6 |
| +Div. HPs | 5 | Deit3b | - | **85.4** | 39.9 | 46.5 | 76.0 | 39.0 | **85.3** | 35.3 | 44.1 | 75.9 | 38.7 | **85.4** | **38.5** | **45.4** | **77.4** | **40.7** |
| DivDis | 5 | Deit3b | IN-R | 85.1 | 36.3 | 41.8 | 77.2 | 40.2 | 84.8 | **40.7** | 42.5 | 76.2 | 38.9 | 85.2 | 35.8 | 40.8 | 77.2 | 40.2 |
| A2D | 5 | Deit3b | IN-R | 85.1 | 37.8 | 45.2 | 77.2 | 40.3 | 84.5 | 39.3 | 45.1 | 75.5 | 39.1 | 85.2 | 36.6 | 44.3 | 77.3 | 40.4 |
| HDR | 5 | Deit3b | $\alpha_n \uparrow$ | 85.3 | **43.0** | **48.7** | 77.3 | 40.7 | **85.3** | 40.3 | **46.1** | 77.3 | 40.6 | 85.1 | 38.3 | 45.3 | 77.2 | 40.4 |
| DE | 50 | Deit3b | - | 85.5 | 38.8 | 45.8 | 75.6 | 38.5 | **85.4** | 37.5 | 45.0 | 75.5 | 38.4 | **85.5** | 38.1 | 45.2 | 75.7 | 38.6 |
| +Div. HPs | 50 | Deit3b | - | 85.5 | 42.5 | 48.5 | **76.0** | 39.0 | **85.4** | 36.4 | 44.8 | **75.9** | 38.8 | 85.5 | 38.5 | 45.6 | **77.5** | **40.8** |
| HDR | 50 | Deit3b | $\alpha_n \uparrow$ | 83.6 | **50.6** | **53.8** | 75.8 | **39.3** | 83.5 | **39.2** | **46.5** | 75.8 | **39.3** | 82.6 | **39.0** | **45.8** | 74.4 | 38.3 |
| DE | 5 | RN18 | - | **69.8** | 0.5 | 20.8 | 51.9 | 14.6 | **69.8** | 0.4 | 19.4 | **51.9** | 14.6 | **69.8** | 0.4 | 19.5 | **51.9** | **14.6** |
| HDR | 5 | RN18 | $\alpha_n \uparrow$ | 69.6 | 0.6 | **20.8** | 51.8 | 14.6 | 69.6 | 0.5 | **19.6** | 51.8 | **14.6** | 69.7 | 0.5 | **19.6** | **51.9** | **14.6** |

Table 2: **OOD generalization of ensembles.** Models are trained on the ImageNet training split. $M$ is the ensemble size. $\mathcal{D}_{\text{div}}$ corresponds to samples on which the respective diversification objectives are applied. $\alpha_n \uparrow$ denotes samples with high $\alpha_n$ values (see § 2.2). "+Div. HPs" stands for deep ensemble diversified via varying hyperparameters during its members' training. $\lambda$ values used in HDR are the following: $10^{-5}$ for IN-A and IN-R, $10^{-3}$ for C-1 and C-5.

**Ensemble aggregation for OOD generalization.** As a means to exploit such robust hypotheses, we consider 3 aggregation strategies. (1) *Oracle selection*: the best-performing individual model is chosen from an ensemble (Pagliardini et al., 2023; Teney et al., 2022a). The final prediction is given by $f(x; \theta^{m^\star})$ where $m^\star := \arg\max_m \text{Acc}(f^m, \mathcal{D}^{\text{ood}})$. (2) *Prediction ensemble* is a vanilla prediction ensemble where the logit values are averaged: $\frac{1}{M}\sum_m f^m(x)$ (Wortsman et al., 2022). (3) *Uniform soup* (Wortsman et al., 2022) averages the weights themselves. The final prediction is given by $f(x; \frac{1}{M}\sum_m \theta^m)$.

**HDR improves OOD generalization for ensembles.** We show the OOD generalization performance of ensembles in Table 2, for the three ensemble prediction aggregation strategies described above. We observe that our framework (HDR) is superior in OOD generalization performance for the prediction ensemble and uniform soup while being on par with best baselines for the oracle selection. HDR is particularly strong in the prediction ensemble (e.g. 48.7% for $M = 5$ and 53.8% for $M = 50$ on ImageNet-R) and uniform soup (e.g. 46.1% for $M = 5$ and 46.5% for $M = 50$ on ImageNet-R). We contend that the increased ensemble diversity contributes to the improvements in OOD generalization. We also remark that the HDR framework (HDR) envelops the performance of A2D and DivDis in this experiment. Together with the superiority of computational efficiency (as discussed at the end of § 3.4) of HDR over the previous diversification methods, this demonstrates that HDR provides a scalable solution for ensemble diversification on ImageNet scale.

**Deep ensembles are a strong baseline.** We also note that deep ensemble, particularly with diverse hyperparameters, provides a strong baseline, outperforming dedicated diversification methodologies under the oracle selection strategy when $M = 5$. It also provides a good balance between ID (ImageNet validation split) and OOD generalization.

## 3.4 OOD DETECTION

We study the impact of ensemble diversification on OOD detection capabilities of an ensemble. Once an ensemble is trained, we compute the epistemic uncertainty, or likelihood of the sample being OOD, following two schemes, $\eta_{\text{BMA}}$ and $\eta_{\text{PDS}}$ introduced in §2.4.

**HDR and PDS together lead to superior OOD detection performance.** We show the OOD detection results in Table 3. We mainly compare PDS to BMA because the latter is considered as the most competitive baseline (Mukhoti et al., 2023) for uncertainty quantification. Comparison to other popular OOD detection baselines (Liu et al., 2020; Xia & Bouganis, 2022) can be seen below the PDS results for Deit3b backbone. Comparison to ResNet18 (He et al., 2016) architecture can be seen the table for Deit3b (see Appendix A.2 for details). For the BMA scores, deep ensemble remains a strong baseline. In particular, when the hyperparameters are varied ("+Diverse HPs"), the detection AUROC reaches the maximal performance among the ensembles using the BMA scores. The quality of PDS is more sensitive to the ensemble diversity, as seen in the jump from the deep ensemble (e.g. 58.9% for OpenImages) to the diverse-HP variant (88.9%). However, when the ensemble is sufficiently diverse, such as when trained with HDR, the PDS leads to high-quality OODness scores. HDR with PDS achieves the best AUROC across the board, including the BMA variants.

**Influence of diversification strength ($\lambda$).** We further study the impact of ensemble diversification on the OOD detection with the PDS estimator. In Figure 4, we observe that strengthening the diversification objective (higher $\lambda$) indeed leads to greater diversity (higher PDS), with a jump at around $\lambda \in [10^{-1}, 10^1]$. This range corresponds to the jump in the OOD detection performance (higher AUROC).

| Method | Unc. score | C-1 | C-5 | iNat | OI |
|---|---|---|---|---|---|
| 1 model | BMA | 61.5 | 83.3 | 95.8 | 90.9 |
| DE | BMA | 61.9 | 83.5 | 95.8 | 91.1 |
| +Div. HPs | BMA | **64.2** | **86.1** | **96.9** | **92.3** |
| DivDis | BMA | 59.8 | 84.3 | 96.6 | 92.2 |
| A2D | BMA | 59.4 | 83.5 | 96.6 | 91.6 |
| HDR | BMA | 64.1 | 84.5 | 96.0 | 91.5 |
| DE | PDS | 56.5 | 62.5 | 59.2 | 58.9 |
| +Div. HPs | PDS | 64.3 | 84.9 | 92.6 | 88.9 |
| DivDis | PDS | 60.0 | 85.1 | 96.9 | 93.9 |
| A2D | PDS | 59.9 | 85.0 | 97.1 | 93.9 |
| HDR | PDS | **68.1** | **89.4** | **97.7** | **94.1** |
| HDR | $\overline{E}(f)$ | 63.3 | 85.8 | **97.7** | 90.8 |
| HDR | $\overline{H}(p)$ | 58.0 | 82.5 | 96.0 | **91.6** |
| HDR | $\overline{p}$ | **67.3** | **87.4** | 80.9 | 82.9 |
| HDR | Ens. $H(p)$ | 58.0 | 82.6 | 96.0 | **91.6** |

(a) Deit3b

| Method | Unc. score | C-1 | C-5 | iNat | OI |
|---|---|---|---|---|---|
| DE | BMA | 66.4 | 86.1 | **86.4** | 80.1 |
| HDR | BMA | **67.8** | **87.8** | 86.2 | **80.2** |
| DE | PDS | 64.4 | 77.4 | 75.0 | 76.1 |
| HDR | PDS | **68.6** | **86.0** | **87.3** | **81.2** |

(b) ResNet-18

Table 3: **OOD detection via ensembles.** For each OOD dataset (IN-C-1, IN-C-5, iNaturalist, and OpenImages), the ensembles are tasked to detect the respective OOD samples among IN-Val samples (ImageNet validation split). We show the AUROC scores for the OOD detection task. Ensemble size is fixed at $M = 5$. "Uncertainty score" refers to the epistemic uncertainty computation framework discussed in §2.4 as well as other methods for OOD detection discussed in Appendix A.3. "+Div. HPs" stands for deep ensemble diversified via varying hyperparameters during its members' training. $\lambda$ values used in HDR are the following: 0.5 for iNat and Oi, 5 for C-1 and C-5.

**Influence of ensemble size.** How ensemble size influences performance of our method? We can see that increasing ensemble size helps to improve AUROC for OOD detection on IN-C-1 (Figure 4). On other datasets increasing ensemble size only marginally helps, but using 5 models provides already a significant improvement over the smallest possible ensemble of size 2. It is also important to mention, that HDR framework is computationally more efficient w.r.t. ensemble size $M$ than for the previous methods such as A2D and DivDis: since we train ensembles for the fixed number of epochs, training complexity for HDR is $O(1)$ thanks to stochastic model pairs selection, while for A2D and DivDis it is $O(M^2)$.

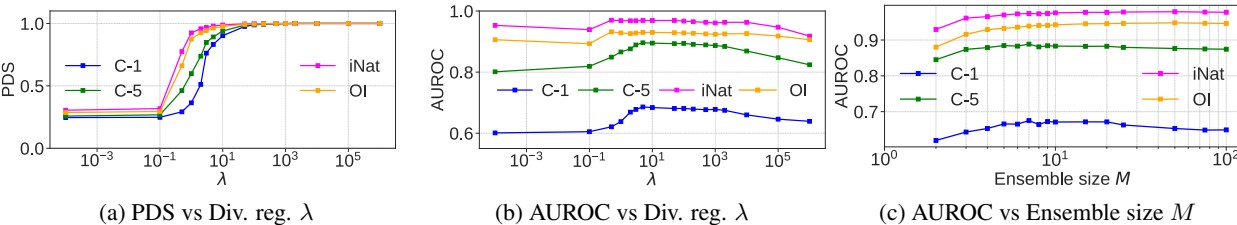

| (a) PDS vs Div. reg. $\lambda$ | (b) AUROC vs Div. reg. $\lambda$ | (c) AUROC vs Ensemble size $M$ |

Figure 4: **Factor analysis for OOD detection**. We show the model answer diversity, measured by PDS, and the OOD detection performance, measured by AUROC, against $\lambda$ values, the loss weight for the disagreement regularizer, and the ensemble size $M$.

## 4 RELATED WORK

**Ensembling** is a well-known technique that aggregates the outputs of multiple models to make more accurate predictions (Breiman, 1996; 2001; Hansen & Salamon, 1990; Ho, 1995; Krogh & Vedelsby, 1994). It was shown that diversity in the outputs of the ensemble members leads to better gains in performance (Krogh & Vedelsby, 1994; Brown et al., 2005; Abe et al., 2022) because they make independent errors (Goodfellow et al., 2016; Hansen & Salamon, 1990).

In addition, it has been shown empirically (Dong et al., 2022) and theoretically (Yong et al., 2024; Hao et al., 2024) that diverse ensembles can also improve OOD generalization.

**Diversity through regularizers.** Various auxiliary training objectives have been proposed to encourage diversity across models' weights (D'Angelo & Fortuin, 2021; de Mathelin et al., 2023a;b; Wang & Ji, 2023), features (Chen et al., 2024; Yashima et al., 2022; Yong et al., 2024), input gradients (Ross et al., 2020; Teney et al., 2022a;b; Trinh et al., 2024), or outputs (D'Angelo & Fortuin, 2021; Lee et al., 2023; Liu & Yao, 1999; Pagliardini et al., 2023; Scimeca et al., 2023). D'Angelo & Fortuin (2021) showed that a regularizer that repulses the ensemble members' weights or outputs leads to ensembles with a better approximation of Bayesian model averaging. This idea was extended by repulsing features (Yashima et al., 2022) and input gradients (Trinh et al., 2024). Since ensemble are most useful when the errors of its members are uncorrelated (Krogh & Vedelsby, 1994), the closest of the above objective is to diversify their outputs. This cannot be guaranteed with other objectives such as weight diversity for example, since two models could implement the exact same function with different weights due to the many symmetries in the parameter space of neural networks. For this reason, this paper focuses on methods for output-space diversification (Lee et al., 2023; Pagliardini et al., 2023). These were also highlighted as state-of-the-art in a recent survey on diversification (Benoit et al., 2024).

**Diversity without modifying the training objective.** The most straightforward way to obtain diverse models is to independently train them with different seeds (Deep Ensembles (Lakshminarayanan et al., 2017) and Bayesian extensions (Wilson & Izmailov, 2020)), hyperparameters (Wenzel et al., 2020), augmentations (Li et al., 2023), or architectures (Zaidi et al., 2021). A computationally cheaper approach is to use models saved at different points during the training (Huang et al., 2017) or models derived from the base model by applying dropout (Gal & Ghahramani, 2016) or masking (Durasov et al., 2021). The "mixture of experts" paradigm (Zhou et al., 2018) can also be viewed as an ensemble where diversification happens by assigning different training samples to different ensemble members. Our experiments use Deep Ensembles (Lakshminarayanan et al., 2017) and ensembles of models trained with different hyperparameters (Wenzel et al., 2020) as baselines since they are strong approaches to OOD detection (Ovadia et al., 2019) and OOD generalization especially when combined with "model soups" (Wortsman et al., 2022).

**Hard samples mining methods for OOD generalization.** Our approach to identifying hard samples in the training set is similar to hard sample mining methods used for worst-group robustness (Liu et al., 2021; Qiu et al., 2023; LaBonte et al., 2023). These methods aim to improve model performance on test samples from minority groups underrepresented in the training set (e.g. photos of animals in unusual contexts, such as a cow on a beach).

While older worst-group robustness methods often required additional training labels for minority groups. The above mentioned approaches address this by upweighting the cross-entropy loss for the samples misclassified by a model preliminary trained with Empirical Risk Minimisation (ERM) on the full training dataset (Liu et al., 2021). Extensions to this work include retraining only the last layer of the model (Qiu et al., 2023) and using disagreements between multiple models in addition to misclassification to identify which samples to up-weight (LaBonte et al., 2023).

Our approach differs. We do not require training a separate model with ERM first. We use hard samples for diversification, not for classification objectives.

## 5 Conclusions

Ensemble diversification has many implications for treating one of the ultimate goals of machine learning, handling out-of-distribution (OOD) samples. Training a large number of diverse hypotheses on a dataset is a way to generate candidates that may have the desired OOD behaviour (i.e. better OOD generalization). And the diversity of hypotheses can help distinguish ID from OOD samples by measuring disagreements across ensemble members. Despite these benefits, diverse-ensemble training has previously remained a lab-bound concept for two reasons. Previous approaches were computationally expensive (scaling quadratically with ensemble size) and required a separate OOD dataset to nurture the diverse hypotheses.

We have addressed these challenges through the novel Hardness-based Diversification Regularizer (HDR). HDR identifies hard samples suitable for disagreement from the training data, bypassing the need to prepare a separate OOD data. HDR also employs a stochastic pair selection to reduce the quadratic complexity of previous approaches to a constant one. We have demonstrated good performance of HDR on OOD generalization and detection tasks, both at the ImageNet scale, a largely underexplored regime in the ensemble diversification community. In particular, for OOD detection, our novel diversity measure of Predictive Diversity Score (PDS) amplifies the benefits of diverse ensembles for OOD detection.

**Limitations.** This work has focused on solving the applicability of disagreement-based diversification on realistic datasets. The contributions are thus mostly in the implementation, and the results focus on empirical benefits. Work is needed to examine theoretical justifications for the method and characterize the exact conditions under which it should provide benefits. Similarly, the proposed PDS is a conceptually sound measure of epistemic uncertainty, but work is also needed to characterize the exact conditions where it is practically superior to alternatives.

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

## A    APPENDICES

### A.1    VARYING THE NUMBER OF TRAINABLE LAYERS

To perform a sensitivity study to the number of layers diversified for each ensemble member we trained only one last layer of DeiT-3b and compared it to the ensemble from the main experiments with the last two layers trained. Both ensembles have size 5 and were trained on the ImageNet training split. The results can be seen in Table 4. Generalization performance did not change much, with the biggest change for ImageNet-C with the corruption strength 5 where ensemble accuracy dropped from $40.8\%$ for one layer to $40.6\%$ for two layers. However, OOD detection performance is better across the board for the case when two layers are diversified, for example, the detection AUROC scores for one layer diversified vs two layers diversified are $0.928$ vs $0.941$ for OpenImages and $0.964$ vs $0.977$ for iNaturalist. We believe that it can be explained by the fact that when one linear layer is trained with cross-entropy loss the optimization problem becomes convex making it harder for disagreement regularizer to promote diversity for different solutions, i.e. ensemble members tend to have similar weight matrices and disagree on OOD samples less.

| | Ensemble Acc. | | | | | AUROC | | | |
|---|---|---|---|---|---|---|---|---|---|
| # Layers | Val | IN-A | IN-R | C-1 | C-5 | C-1 | C-5 | iNat | OI |
| 1 | 85.2 | 42.3 | **48.2** | **77.3** | **40.8** | 67.7 | 88.9 | 96.4 | 92.8 |
| 2 | **85.3** | **42.4** | 48.1 | **77.3** | 40.6 | **68.1** | **89.4** | **97.7** | **94.1** |

Table 4: Varying the number of trainable layers.

### A.2    RESNET18 RESULTS

To check the applicability of our method to other architectures we trained an ensemble of 5 models with the whole model but last layer frozen using ResNet18 as a feature extractor. We compared HDR with A2D disagreement regularizer and stochastic sum size $|\mathcal{I}| = 2$ vs deep ensemble in Table 5. Both ensembles were trained on the ImageNet training split. Deep ensemble and HDR-A2D have similar generalization performance, with the biggest difference for ImageNet-C with the corruption strength 1 where ensemble accuracy dropped from $51.9\%$ for deep ensemble to $51.8\%$ for HDR-A2D. Nevertheless, HDR-A2D shows better OOD detection performance across the board, for example, the detection AUROC scores for one deep ensemble vs HDR-A2D are $0.802$ vs $0.812$ for OpenImages and $0.865$ vs $0.973$ for iNaturalist. Ensemble accuracy on ImageNet-A is less than $1\%$ for both ensembles: $0.5\%$ and $0.6\%$ because this dataset was created with a goal to minimize ResNet performance on it.

| | Ensemble Acc. | | | | | AUROC | | | |
|---|---|---|---|---|---|---|---|---|---|
| Method | Val | IN-A | IN-R | C-1 | C-5 | C-1 | C-5 | iNat | OI |
| Deep Ensemble | **69.8** | 0.5 | **20.8** | **51.9** | **14.6** | 67.0 | 86.9 | 86.5 | 80.2 |
| HDR | 69.6 | 0.6 | **20.8** | 51.8 | **14.6** | **68.6** | **87.9** | **87.3** | **81.2** |

Table 5: With a ResNet18 backbone.

### A.3    OTHER UNCERTAINTY SCORES

In this section, we define the uncertainty scores used for comparison in Table 3.

**Average Energy $(\overline{E}(f))$**    We compute the energy uncertainty score (Liu et al., 2020) for each ensemble member and then average energy values among ensemble members (we omit the temperature term $T$ from the original definition by setting it always equal to 1):

$$\overline{E}(f) = -\frac{1}{M}\sum_{m=1}^{M}\log\sum_{c=1}^{C}e^{f_c^m(\mathbf{x})} \tag{11}$$

**Average Entropy ($\overline{H}(p)$) and Ensemble Entropy (Ens. $H(p)$):**

$$\overline{H}(p) = \frac{1}{M} \sum_{m=1}^{M} \mathcal{H}\left[p^m(x)\right] \tag{12}$$

$$\text{Ens. } H(p) = \mathcal{H}\left[\frac{1}{M} \sum_{m=1}^{M} p^m(x)\right], \tag{13}$$

where $\mathcal{H}[p(x)] = -\frac{1}{C} \sum_{c=1}^{C} p_c(x) \log p_c(x)$

**Average confidence of ensemble members ($\overline{p}$)** :

$$\overline{p} = \frac{1}{M} \sum_{m=1}^{M} \max_c p_c^m(x) \tag{14}$$

## A.4  COMPARISON TO A TWO-STAGE APPROACH

To perform an ablation study on the way samples for disagreement are selected in Table 6 we compared an ensemble trained with Equation 6 (called "joint" in the table) against a 2-stage approach. Instead of disagreeing on all samples with adaptive weight $\alpha_n$ as in Equation 6 we first computed the confidence of the pre-trained DeiT-3B model on all samples in ImageNet training split and then selected samples with a confidence lower than $0.2$ which resulted in $18002$ samples (to approximately match the sizes of ImageNet-A and ImageNet-R). Then we trained an ensemble by minimizing A2D disagreement regularizer on these samples while minimizing cross-entropy on all other samples. Both ensembles had size 5 and stochastic sum size $|\mathcal{I}| = 2$. While such an approach might sound simpler, HDR is more straightforward and efficient, since there is no need to train an initial model to determine samples for disagreement. Both ensembles have a similar generalization performance, with the biggest difference for ImageNet-R where ensemble accuracy dropped from $48.5\%$ for 2-stage approach to $48.1\%$ for the joint. In contrast, OOD detection performance is significantly better across the board for the joint approach, for example, the detection AUROC scores are $0.845$ vs $0.896$ for ImageNet-C with corruption strength 5 and $0.911$ vs $0.941$ for OpenImages. We think that such a drastic difference in OOD detection performance can be caused by the fact that the set of samples selected for disagreement may be suboptimal which makes the confidence threshold (set as $0.2$ for this experiment) an important hyperparameter and adds even more complexity to the 2-stage approach.

| Type | Val | Ensemble Acc. IN-A | IN-R | C-1 | C-5 | AUROC C-1 | C-5 | iNat | OI |
|------|-----|------|------|------|------|------|------|------|------|
| 2-stage | 85.2 | **42.4** | **48.5** | **77.3** | **40.7** | 59.7 | 84.5 | 96.0 | 91.1 |
| Joint | **85.3** | **42.4** | 48.1 | **77.3** | 40.6 | **68.1** | **89.4** | **97.7** | **94.1** |

Table 6: Comparison with a two-stage approach.

## A.5  SMALL-SCALE EXPERIMENTS

To check the performance of our method on the small-scale datasets, we conducted additional experiments on the Waterbirds dataset (Sagawa* et al., 2020) (Table 7) and DomainNet (Table 8) (Neyshabur et al., 2020). For these experiments we used ImageNet-pretrained ResNet-50 (He et al., 2016) as a backbone architecture. No layers were frozen during training.

**Waterbirds** We use this dataset to compare HDR to A2D and DivDis on a small scale dataset since both papers provided results for it. We report the worst group (test) accuracy for ensembles of size 4. We trained A2D, DivDis, and an

ensemble with HDR and A2D disagreement regularizer on Waterbirds training split. We did not use stochastic sum for HDR-A2D to factor out its influence. A2D and DivDis used the validation set for disagreement. While DivDis discovers a better single model having best accuracy of 87.2% against 83.2% for the proposed HDR-A2D method, the ensemble is clearly better with HDR-A2D: 80.6% vs 78.3% for DivDis.

|       | Oracle selection | Ensemble |
|-------|------------------|----------|
| ERM   | 76.5             | 72.0     |
| DivDis | **87.2**        | 78.3     |
| A2D   | 78.3             | 78.3     |
| HDR   | 83.5             | **80.6** |

Table 7: Worst group test accuracy on Waterbirds

**DomainNet** We use this dataset because it is a popular OOD generalization benchmark. Following the original procedure (Neyshabur et al., 2020) for each column in the table we train the models on all domains except for the test one and report test accuracy on the latter. All ensembles are of size 6. As we can see in Table 8 HDR and Deep Ensemble have the same performance of 58.4% on "Real" test set with HDR being better than Deep Ensemble on all other test sets (e.g. 61.2% vs 58.2% on "Clip" or 53.1% vs 50.9% on "Sketch") as well as in average performance (42.6% vs 41.4%).

In addition to performance comparison, we conducted a sensitivity study to analyze the influence of the $\lambda$ parameter from Equation 6 on ensemble accuracy in OOD generalization tasks on DomainNet (Figures 5 - 10). There is no one single best $\lambda$ value for all test sets, to maximize test performance these values should be selected separately for each test set e.g. $10^{-4}$ for "Clip" or $10^{-5}$ for "Info".

### A.6 OOD DATASETS FOR DISAGREEMENT

To analyze the influence of OOD data used for disagreement we performed additional experiments with ensemble members disagreeing on ImageNet-R and ImageNet-A in Table 9. We compare A2D and Div (Lee et al., 2023) diversification regularizers. Usage of ImageNet-A or ImageNet-R resulted in almost identical (identical after rounding) OOD generalization performance for A2D disagreement regularizer, while for Div regularizer ensemble accuracy on ImageNet-R dropped from 45.2% when using ImageNet-A for disagreement to 41.8% when using ImageNet-R for disagreement. OOD detection performance also does not change much for any combination of regularizer and dataset used for disagreement with the biggest difference in detection AUROC scores 0.973 for Div regularizer and and ImageNet-A disagreement dataset vs 0.969 for Div regularizer and ImageNet-R disagreement dataset.

### A.7 VARIATIONS OF THE STOCHASTIC SUM SIZE

We performed an additional evaluation (Table 10) that shows the benefit of controlling the stochastic sum size ($|\mathcal{I}|$) on the speed of training an ensemble. For example, to train an ensemble of size 5, the time required for 1 epoch grows from 53s for $|\mathcal{I}| = 2$ to 585s for $|\mathcal{I}| = 5$ (without stochastic sum). We could not train an ensemble of 50 models without stochastic sum with our resources, but it already requires 7244s for $|\mathcal{I}| = 10$ vs 2189s for $|\mathcal{I}| = 2$. Standard deviations of training epoch times are computed across 10 different epochs. The speed up is especially important for training an ensemble with 50 models. Since the number of model pairs grows from 1 to $C_{50}^2 = 1125$ in that case, a theoretical time for 1 epoch would be approximately 1225 times higher than the training time for $|\mathcal{I}| = 2$, i.e $\approx 0.5 \cdot 1125 = 663$ GPU hours. An important note here is that training time is affected by moving data between CPU and GPU (only the models

|               | Clip | Info | Paint | Real | Quick | Sketch | Average |
|---------------|------|------|-------|------|-------|--------|---------|
| Deep Ensemble | 58.2 | 18.4 | 47.7  | **58.4** | 14.9 | 50.9 | 41.4 |
| HDR           | **61.2** | **18.8** | **48.4** | **58.4** | **15.7** | **53.1** | **42.6** |

Table 8: Results on DomainNet. $\lambda$ values used in HDR are the following: $10^{-4}$ for "Clip" and "Quick", $10^{-5}$ for "Info", $10^{-6}$ for "Paint" and "Sketch".

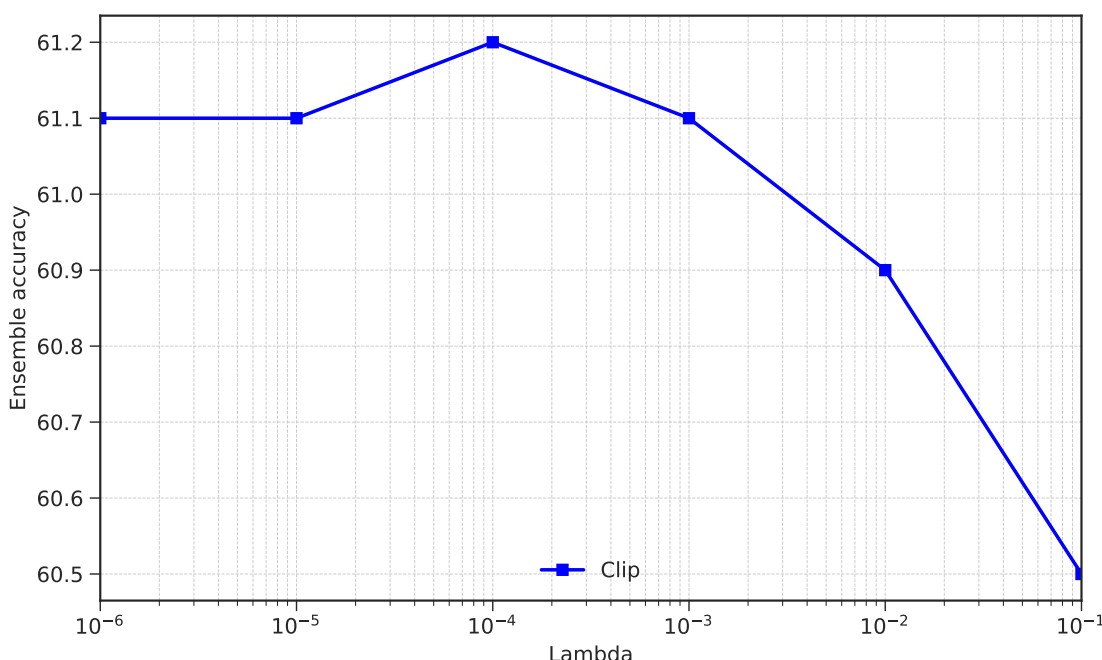

Figure 5: Ensemble accuracy on "Clip" test set against the loss weight for the disagreement regularizer values $\lambda$.

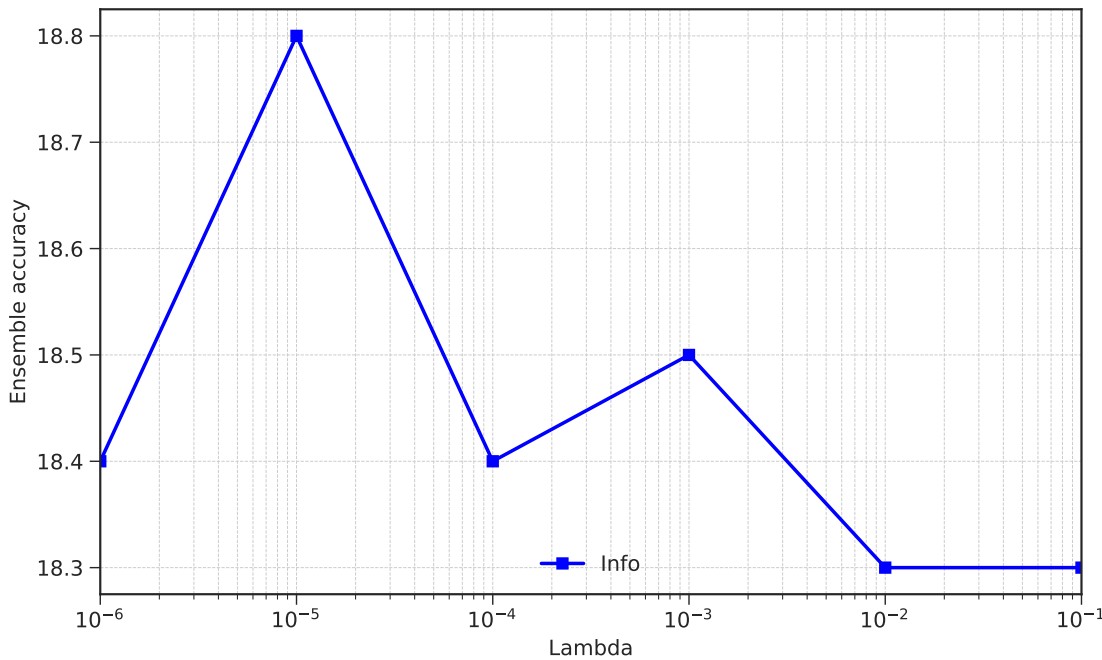

Figure 6: Ensemble accuracy on "Info" test set against the loss weight for the disagreement regularizer values $\lambda$.

used for loss computation are loaded to GPU in our implementation), therefore, it is hard to accurately predict epoch times.

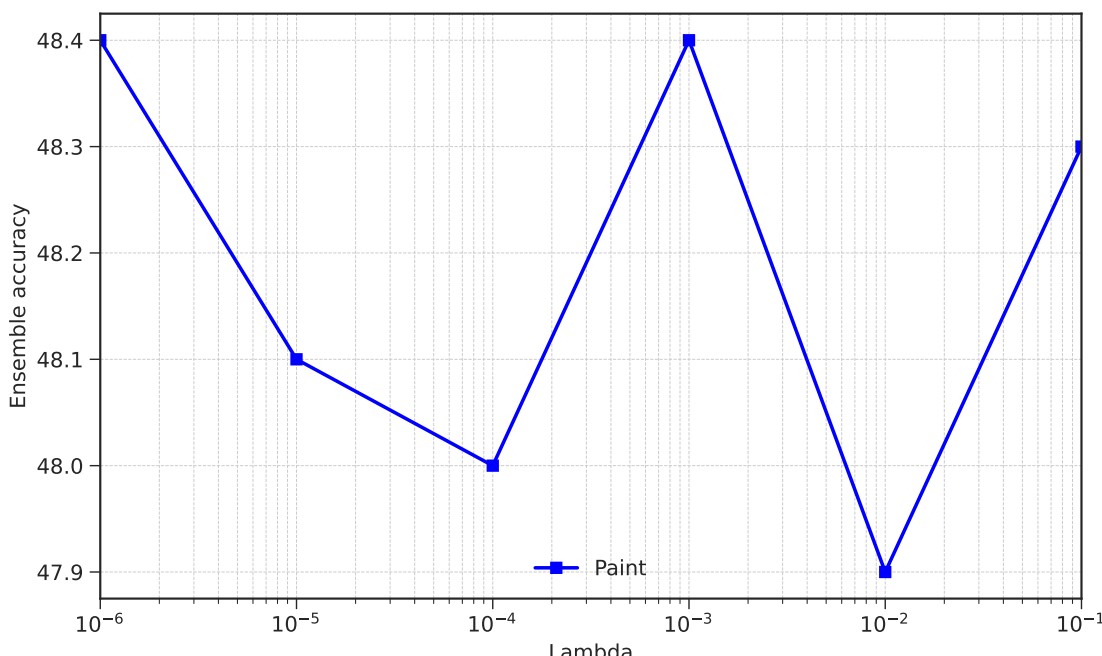

Figure 7: Ensemble accuracy on "Paint" test set against the loss weight for the disagreement regularizer values $\lambda$.

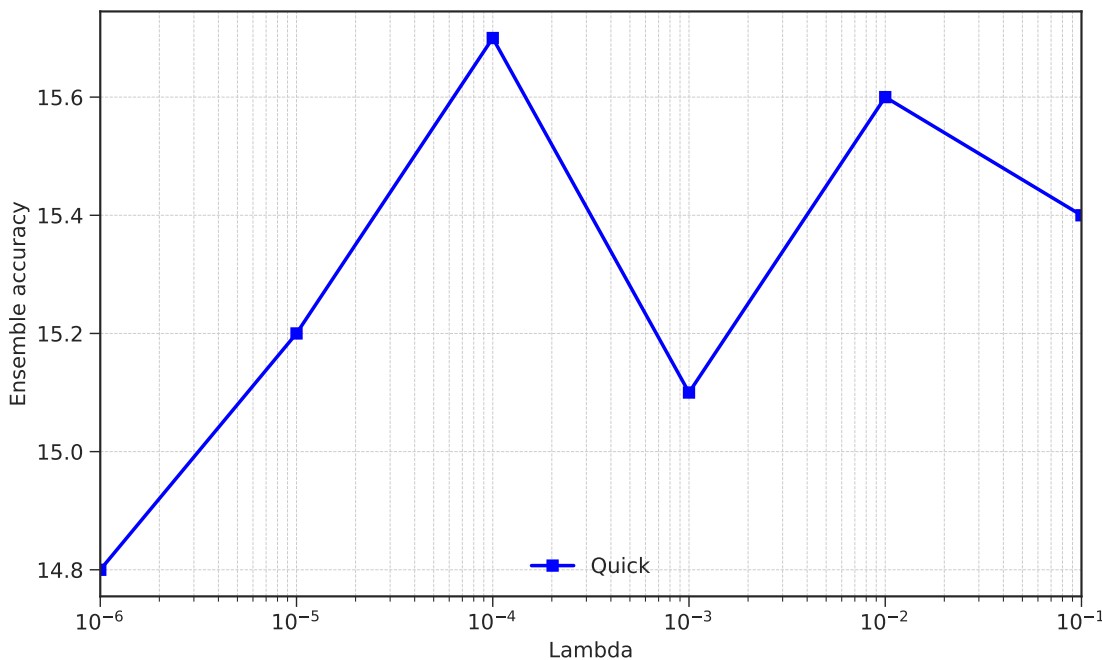

Figure 8: Ensemble accuracy on "Quick" test set against the loss weight for the disagreement regularizer values $\lambda$.

## A.8 JUSTIFICATION FOR $\alpha_n$

In this section we take a deeper look into gradients of the $\mathcal{L}_{\text{HDR}}$ (Equation 6) and justify why regularization weight $\alpha_n$ should depend on the sample-wise cross-entropy loss and scaled down by squared average cross-entropy loss in batch.

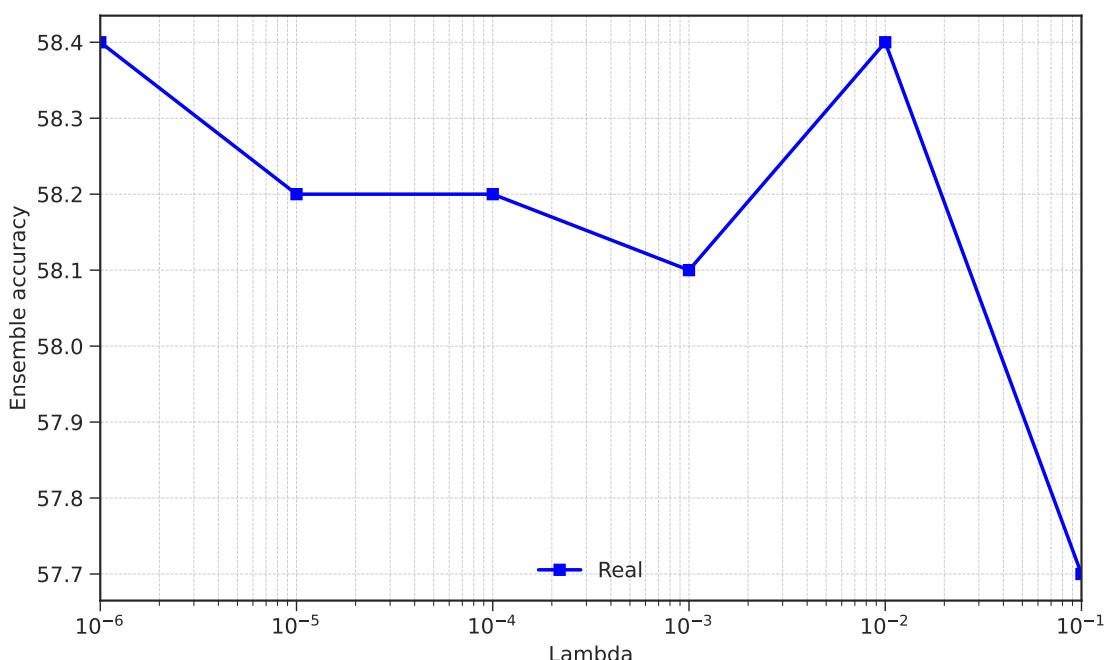

Figure 9: Ensemble accuracy on "Real" test set against the loss weight for the disagreement regularizer values $\lambda$.

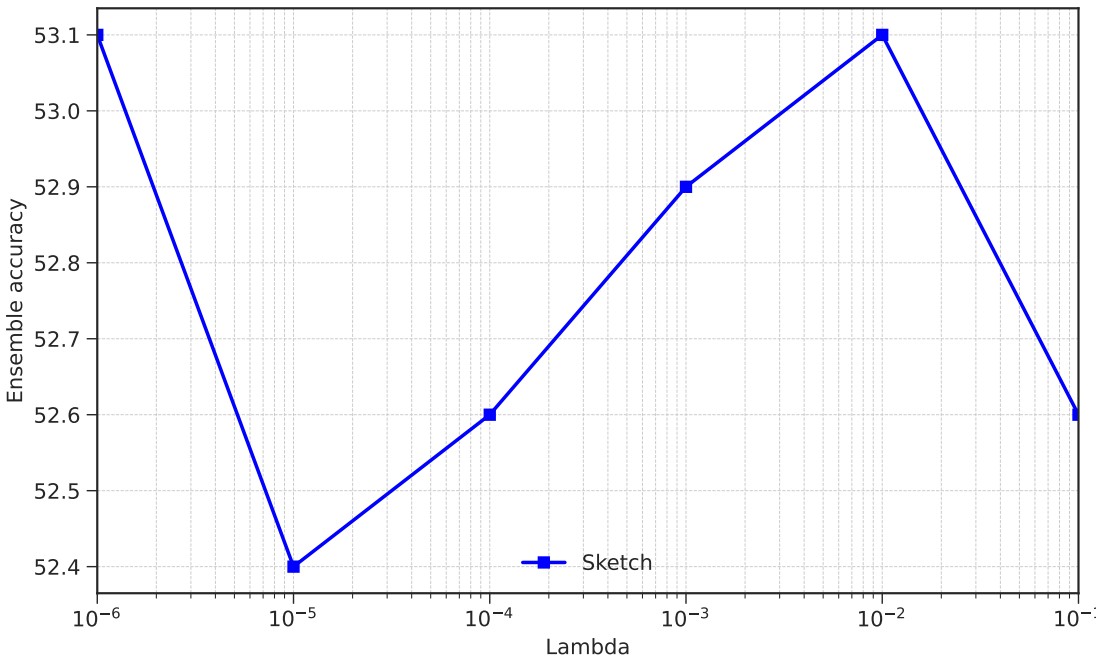

Figure 10: Ensemble accuracy on "Sketch" test set against the loss weight for the disagreement regularizer values $\lambda$.

For the simplicity, we assume that ensemble contains only two models. For some fixed input $x$ with ground truth label $y$ we denote output probabilities as $f = p^1(x)$ and $g = p^2(x)$ for the first and second model correspondingly, while denoting their predictions as $\hat{f} = \mathrm{argmax}_k f_k$ and $\hat{g} = \mathrm{argmax}_k g_k$. We also omit the subscript of $\alpha_n$ for brevity and simply use $\alpha$ instead. In this case, the total training loss on a sample $(x, y)$ has the form:

|  |  | Ensemble Acc. | | | | | AUROC | | | |
| Method | OOD | Val | IN-A | IN-R | C-1 | C-5 | C-1 | C-5 | iNat | OI |
|---|---|---|---|---|---|---|---|---|---|---|
| A2D | IN-A | **85.1** | **37.8** | **45.2** | **77.2** | **40.3** | 59.9 | **85.0** | 97.1 | 93.6 |
| A2D | IN-R | **85.1** | **37.8** | **45.2** | **77.2** | **40.3** | 59.9 | **85.0** | 97.1 | **93.9** |
| Div | IN-A | **85.1** | **37.8** | **45.2** | **77.2** | **40.3** | 59.9 | **85.0** | **97.3** | 93.7 |
| Div | IN-R | **85.1** | 35.7 | 41.8 | **77.2** | 40.2 | **60.0** | **85.0** | 96.9 | 93.8 |

Table 9: OOD Datasets for disagreement.

| M | I | Epoch, s | Val | IN-A | IN-R | C-1 | C-5 | C-1 | C-5 | iNat | OI |
|---|---|---|---|---|---|---|---|---|---|---|---|
| 5 | 2 | **53 ± 5** | **85.3** | **42.4** | **48.1** | **77.3** | **40.6** | 68.6 | 89.6 | **97.7** | 94.1 |
| 5 | 3 | 388 ± 28 | 85.2 | 41.4 | 47.4 | 77.2 | 40.5 | 68.2 | 89.2 | 97.5 | 93.9 |
| 5 | 4 | 423 ± 3 | 85.2 | 40.3 | 46.8 | 77.1 | 40.4 | 70.3 | 89.8 | 97.3 | 94.0 |
| 5 | 5 | 585 ± 111 | 85.1 | 37.6 | 44.9 | 77.0 | 40.2 | **71.1** | **90.3** | 97.0 | 93.7 |
| 50 | 2 | **2189 ± 86** | **83.7** | **50.1** | **54.0** | **75.9** | **39.4** | **60.0** | 82.4 | 93.4 | 87.8 |
| 50 | 5 | 4213 ± 5 | 83.6 | 49.2 | 53.4 | 75.8 | 39.2 | 59.8 | 82.7 | 94.2 | 89.2 |
| 50 | 10 | 7244 ± 27 | 83.4 | 48.5 | 53.0 | 75.6 | 39.1 | 59.7 | **82.8** | **94.5** | **89.6** |

Table 10: Variations of the stochastic sum size.

$$\mathcal{L}_{\text{HDR}}(x,y) := -\log f_y - \log g_y - \alpha \log(f_{\hat{f}}(1 - g_{\hat{f}}) + g_{\hat{f}}(1 - f_{\hat{f}})) - \alpha \log(g_{\hat{g}}(1 - f_{\hat{g}}) + f_{\hat{g}}(1 - g_{\hat{g}})) \quad (15)$$

We can compute its partial derivatives (gradients) w.r.t. $f_y, f_{\hat{f}}, f_{\hat{g}}$ - probabilities predicted by model $f$ for the ground truth, for the prediction of model $f$ and for the prediction of model $g$ correspondingly:

$$\nabla_{f_y}\mathcal{L}_{\text{HDR}}(x,y) = -\frac{1}{f_y} \quad (16)$$

$$\nabla_{f_{\hat{f}}}\mathcal{L}_{\text{HDR}}(x,y) = \frac{\alpha(2g_{\hat{f}} - 1)}{f_{\hat{f}} + g_{\hat{f}} - 2f_{\hat{f}}g_{\hat{f}}} \quad (17)$$

$$\nabla_{f_{\hat{g}}}\mathcal{L}_{\text{HDR}}(x,y) = \frac{\alpha(2g_{\hat{g}} - 1)}{f_{\hat{g}} + g_{\hat{g}} - 2f_{\hat{g}}g_{\hat{g}}} \quad (18)$$

Similar partial derivatives can be obtained w.r.t. $g_y, g_{\hat{f}}, g_{\hat{g}}$.

Since $0 \le C(f,g) = f_y + g_y - 2f_yg_y \le 1$ for $0 \le f_y \le 1, 0 \le g_y \le 1$ (by examining critical and border points we can see that its minimum value is 0 and maximum value is 1 on this square), when $y \ne \hat{f} \ne \hat{g}$, the sign of the derivatives above depends only on the outputs of a single model (either $f$ or $g$). However, when $y = \hat{f}$ or $y = \hat{g}$ gradients start to clash with each other, i.e. the total gradient is obtained by summing two gradients with possibly opposite signs.

Let's consider the case $y = \hat{f}$:

$$\nabla_{f_y}\mathcal{L}_{\text{HDR}}(x,y) = -\frac{1}{f_y} + \frac{\alpha(2g_y - 1)}{C(f,g)} \quad (19)$$

It has two terms: the first term, $\frac{1}{f_y}$ has constant sign, while the sign of the second term, $\frac{\alpha(2g_y - 1)}{C(f,g)}$ depends on the value of $g_y$. This might lead to instabilities in training because the sign of the total gradient can flip during training depending on the current value of $g_y$.

To avoid such instabilities in gradient sign, we make weight $\alpha$ adaptive to the type of sample on which gradient is computed. For easy samples on which model makes correct predictions, i.e. high $f_y$, near-zero gradient value is desirable because we want to keep the prediction for such samples correct. For hard samples, i.e. with low $f_y$, we want the gradient to be dominated by the second term that is responsible for models disagreement. Therefore, we make $\alpha$ inversely proportional to $f_y$ (to be precise we make it proportional to $-\log f_y$ for computational stability reasons).

The need for inverse proportion can be seen after checking when absolute values of the two gradient terms equal to each other:

$$\frac{1}{f_y} = \frac{|-1|}{|f_y|} = \frac{\alpha|2g_y - 1|}{C(f,g)} \tag{20}$$

$$f_y = \frac{C(f,g)}{\alpha|2g_y - 1|} \tag{21}$$

If we set $\alpha$ to some constant value $\overline{\alpha}$, there will always be a value of $\overline{f_y} = \frac{C(f_y, g_y)}{\overline{\alpha}|2g_y - 1|}$, such that for $f_y(x) < \overline{f_y}$ the first term dominates the gradient and for $f_y > \overline{f_y(x)}$ the second term dominates the gradient. Such behavior will again lead to clashes between the terms depending on the value of $f_y$.

The only way to avoid such clashes is to set $\alpha$ proportional to $\frac{1}{f_y}$, i.e. $\alpha = \gamma\frac{1}{f_y}$, for some $\gamma > 0$. Then from Equation 20 we will get the following condition for the second term dominance in the total gradient (for $0 \leq C(f,g) \leq 1$):

$$\frac{1}{f_y} \leq \frac{\gamma|2g_y - 1|}{f_y C(f,g)} \tag{22}$$

$$\gamma^{-1} \leq |2g_y - 1| \leq \frac{|2g_y - 1|}{C(f,g)} \tag{23}$$

However, making $\alpha$ inversely proportional to $f_y$ is not enough, as in the beginning of the training when $f_y$ is small on all training samples, the second term always dominates the gradient in Equation 19 resulting only in outputs diversification and neglecting the classification task. To solve this problem, we scale down $\alpha$ by a squared average cross-entropy loss in batch as shown in Equation 5, the square is important to keep the average value of $\alpha$ dependent on the average cross entropy as explained in Section 2.2.

Similar reasoning can be applied to the cases, when $y = \hat{g}$ or $y = \hat{f} = \hat{g}$. The argument holds for gradients computed w.r.t. $g_y, g_{\hat{f}}, g_{\hat{g}}$ and scenarios with more than two models in the ensemble.

A.9    DIFFERENT WAYS TO COMBINE LOSSES

As we have seen in § A.8 the classification and diversification objectives may clash. Let's consider the simplified form of Equation 6 for sample $n$ and model pair $(m, l)$: $L_{\text{HDR}} = L_{main} + \alpha_n \cdot L_{div}$. When $\alpha_n > 0$, both terms are applied to the same sample $n$, leading to potential clash in objectives. By default we control the relative importance of the terms through $\alpha_n$: for harder samples, we make $\alpha_n$ greater, such that the relative weight of the diversification term is greater and vice versa.

Another option is to control the weights in the "convex sum" way: $L_{\text{HDR}} = (1 - \alpha_n)L_{main} + \alpha_n L_{div}$.

We compared both options in Table 11. They have almost identical results in OOD generalization on IN-Val and IN-C-1/5 while on IN-A and IN-R our default way to combine weights performs better (42.4 vs 40.9 and 48.1 vs 47.4 correspondingly). For OOD detection default approach is better across the board, e.g. 89.4 vs 82.5 for IN-C-1.

| | Ensemble Acc. | | | | | AUROC | | | |
|---|---|---|---|---|---|---|---|---|---|
| Losses Combining | Val | IN-A | IN-R | C-1 | C-5 | C-1 | C-5 | iNat | OI |
| Convex sum | **85.4** | 40.9 | 47.4 | **77.4** | **40.8** | 59,6 | 82.5 | 95.5 | 90.7 |
| Default | 85.3 | **42.4** | **48.1** | 77.3 | 40.6 | **68.1** | **89.4** | **97.7** | **94.1** |

Table 11: Compare different ways to combine losses.

### A.10 SAMPLES' HARDNESS WEIGHT DYNAMICS

To observe which data points are identified as hard samples by the models and how this is connected to quantitative performance we conducted a fluctuation analysis for an ensemble at the beginning and at the end of training. For that reason, we recorded $\alpha_n$ (Equation 6) values throughout the training, sorted them according to their magnitude, and grouped into 100 bins, so that the biggest weights corresponded to bin 0, while the smallest to bin 99. For each bin we computed the *fluctuation ratio*, i.e. the ratio of fluctuating samples per bin. We considered a sample as fluctuating if it changed relative position with respect to the median weight value among all sample-wise weights for the current epoch. For example, if during the current epoch, a sample's weight is higher than the median value and during the next epoch it is lower than the median value, then we call such sample as a *fluctuating sample*.

The fluctuation analysis revealed that approximately half of the samples fluctuate during the early stages of training (fluctuation ratio is around $0.5$ for all bins during the first epoch in Figure 11) when ensemble is undertrained and its performance is lower than during the later epochs. As training progresses, the models gradually converge on a more stable in comparison to the early stages of training set of hard samples (when comparing the first epoch to the last epoch fluctuation ratio drops from $0.5$ to $0.3$ for the samples from the first 15 bins, i.e. the hardest samples as can be seen in Figure 11). In addition to that we notice that average weights magnitude per bin stabilises after a few epochs (when comparing the first and the last epochs difference between weights magnitude is between 5 and 0.5 depending on the bin as shown in Figure 12; when comparing the last and the second last epoch difference is almost zero for all bins as shown in Figure 13) with samples from bins between 0 and 14 (the hardest samples) having noticeably higher weights than the samples from the other bins (during the last or the second last epochs weights are between 1 and 5 for them, while for other bins they are below 1 in Figure 13). The latter is not the case during the first epoch when average weight magnitude is uniformly low across all the bins (during the first epoch average weight magnitude is around $0.15$ for all the bins in Figure 12).

For this experiment, we trained an ensemble of size 5 with $\lambda$ equals $0.5$ and frozen Deit3b backbone for 10 epochs (the same ensemble was used for OOD detection on iNaturalist and OpenImages in Table 3).

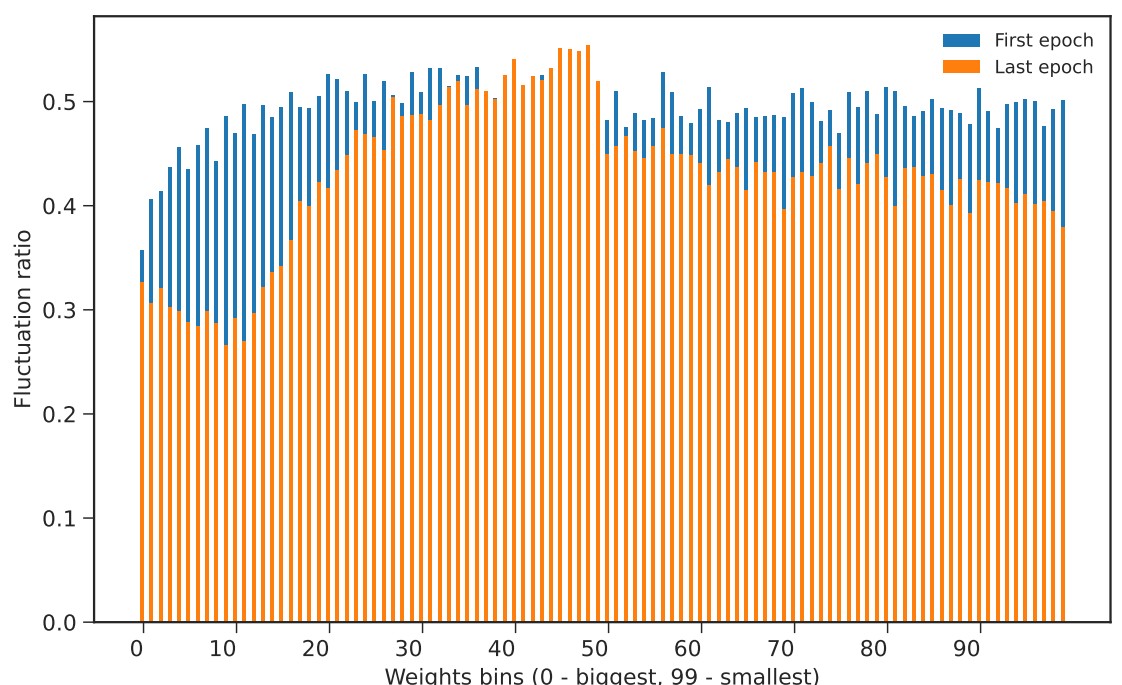

Figure 11: Fluctuations ratio for different weights bins.

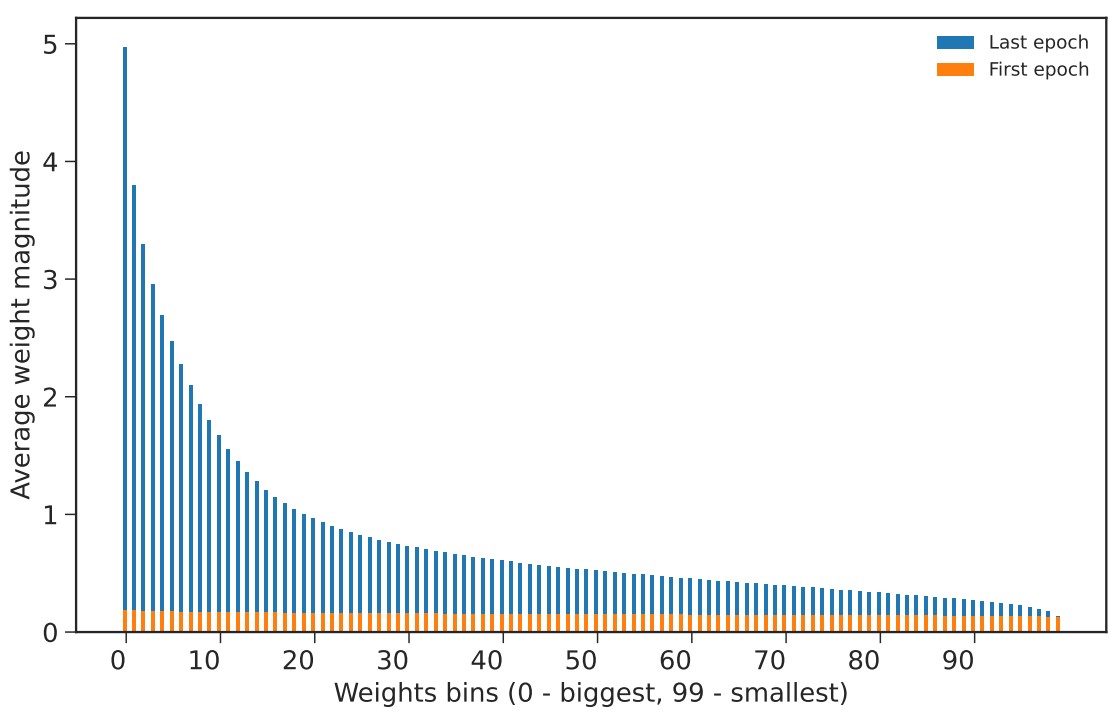

Figure 12: Average weights for different weight bins during the first and the last epochs.

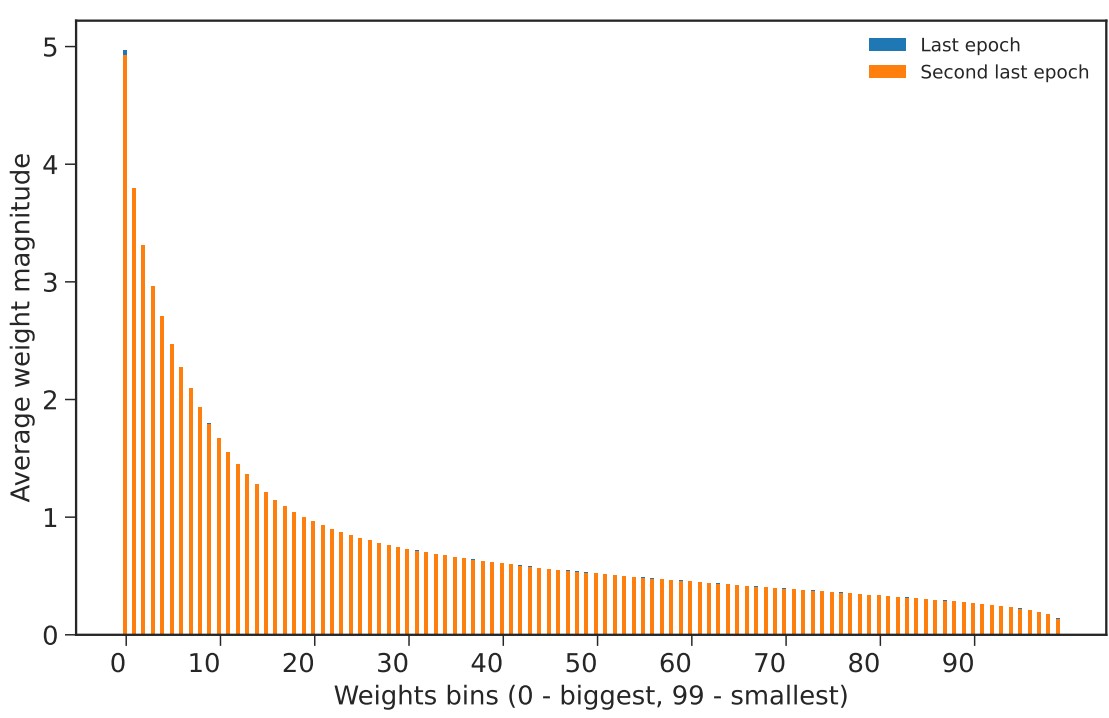

Figure 13: Average weights for different weight bins during the last two epochs.

