# OpenReview forum: "Scalable Ensemble Diversification for OOD Generalization and Detection"
_ICLR.cc/2025/Conference — Submitted to ICLR 2025_

### Official Review · Reviewer_eQEt · 2024-11-01

**Soundness:** 3
**Presentation:** 3
**Contribution:** 3
**Rating:** 5
**Confidence:** 3

**Summary:**

This paper proposes a new ensemble framework where model prediction diversity is encouraged on only hard samples. They also propose a new diversity measurement for OOD detection. The experiments show that the proposed method can effectively enforce the prediction diversity of the ensembled models, which lead to a better OOD performance.

**Strengths:**

1) The paper gives a reasonable and easy-to-understand motivation to encourage diversity on hard samples for single domain generalization.
 2) Extensive experiments on 2 different tasks are conducted to demonstrate the effectiveness of the proposed method.

**Weaknesses:**

1) It would be better to see some performance results on more widely used public benchmark datasets for single domain generalization, such as PACS, DomainNet, so that the proposed method can be compared with more baseline methods.
2) The overall performance improvement of the proposed method seems to be marginal on 3 out of 5 of the domains in Table 2. Can you provide more analysis on the reason behind these results?

**Questions:**

Please refer to the Weakness section.

---

> ### Author Response · Authors · 2024-11-18
> **Thank you + W2. The overall performance improvement of the proposed method seems to be marginal on 3 out of 5 of the domains in Table 2. Can you provide more analysis on the reason behind these results?**
>
> We thank the reviewer for writing the review. Below we will provide responses to the weaknesses (response to the weakness 1 will be added later once experiments are finished).
>
> ## W2. The overall performance improvement of the proposed method seems to be marginal on 3 out of 5 of the domains in Table 2. Can you provide more analysis on the reason behind these results?
>
> The primary objective of this paper is to introduce a methodology aimed at improving ensemble diversity.
>
> Our results successfully demonstrate that HDR enhances ensemble diversity, as shown in Table 1.
>
> OOD generalization represents a common application scenario in the field [a, b, c, d, e]. The results in Table 2 verify the usual wisdom that diversification helps OOD generalisation. While the improvement is not particularly pronounced, our findings suggest that the diversified ensemble remains a viable alternative to other ensemble methods, such as DE.
>
> For OOD detection, however, diversification proves to be highly beneficial (see Table 3), achieving state-of-the-art performance across various benchmarks.
>
> We believe the effectiveness of HDR is sufficiently demonstrated and that the contribution is significant enough for acceptance.
>
> [a] Yoonho Lee, Huaxiu Yao, and Chelsea Finn. Diversify and disambiguate: Out-of-distribution robustness via disagreement. In
> International Conference on Learning Representations, 2023.
>
> [b] Matteo Pagliardini, Martin Jaggi, François Fleuret, and Sai Praneeth Karimireddy. Agree to disagree: Diversity through disagreement
> for better transferability. In International Conference on Learning Representations, 2023.
>
> [c] Teney, D., Abbasnejad, E., Lucey, S., & Hengel, A.V. (2021). Evading the Simplicity Bias: Training a Diverse Set of Models Discovers Solutions with Superior OOD Generalization. 2022 IEEE/CVF Conference on Computer Vision and Pattern Recognition (CVPR), 16740-16751.
>
> [d] Ross, A.S., Pan, W., Celi, L.A., & Doshi-Velez, F. (2019). Ensembles of Locally Independent Prediction Models. AAAI Conference on Artificial Intelligence.
>
> [e] Teney, D., Peyrard, M., & Abbasnejad, E. (2022). Predicting is not Understanding: Recognizing and Addressing Underspecification in Machine Learning. European Conference on Computer Vision.

---

> > ### Author Response · Authors · 2024-11-21
> > **W1. Results on DomainNet and PACS**
> >
> > We thank the reviewer for the suggestion and will provide the results on DomainNet and PACs before 26.11

---

> > > ### Author Response · Authors · 2024-11-26
> > > **W1. Results on DomainNet and PACS vol. 2**
> > >
> > > We have conducted the experiments on DomainNet and added their results to the paragraph “DomainNet” in §A.5 as well as in Table 8 in the updated PDF.
> > >
> > > Here is a copy of that table:
> > >
> > > |               Method        | Clip  | Info  | Paint | Real  | Quick | Sketch | Average |
> > > |-----------------------------|-------|-------|-------|-------|-------|--------|---------|
> > > | Deep Ensemble              |  58.2 |  18.4 |  47.7 | **58.4** |  14.9 |  50.9   |   41.4  |
> > > | HDR             | **61.2** | **18.8** | **48.4** | **58.4** | **15.7** | **53.1** | **42.6** |
> > >
> > > HDR and Deep Ensemble have the same performance of $58.4\%$ on "Real" test set with HDR being better than Deep Ensemble on all other test sets (e.g. $61.2\%$ vs $58.2\%$ on "Clip" or $53.1\%$ vs $50.9\%$ on "Sketch") as well as in average performance ($42.6\%$ vs $41.4\%$).
> > >
> > > All ensemble training time was constrained to 1 hour per model (to fit in the rebuttal time budget); instead of the best checkpoint based on test accuracy, we used the latest checkpoint for evaluation (to reflect the realistic scenario where test datasets are not available). That resulted in performances lower than state-of-the-art but sufficient to compare HDR and deep ensemble.
> > >
> > > We have not computed results on PACS dataset due to time constraints. We believe that the trend of HDR surpassing Deep Ensemble performance is clear from DomainNet.
> > >
> > > Nevertheless, we can run the experiments on PACS during the extended discussion period if the reviewer finds it necessary.

---

> ### Author Response · Authors · 2024-12-01
> **Thank you for your feedback!**
>
> Dear reviewer eQEt, we would be grateful for an updated score if you have no further concerns. However, if you have any outstanding concerns, we would appreciate you sharing them now so that we have sufficient time to ensure the paper meets your expectations.

---

### Official Review · Reviewer_sLK4 · 2024-11-04

**Soundness:** 3
**Presentation:** 2
**Contribution:** 2
**Rating:** 5
**Confidence:** 4

**Summary:**

This paper tackles the out-of-distribution (OOD) generalization and detection via the ensemble diversification approach. Specifically, the authors propose the Hardness-based Diversification Regularizer (HDR) to identify OOD samples from the training data and make disagreement between the members of the ensemble, which is different from existing methods requiring additional unlabeled OOD data. Furthermore, this paper proposes a new OOD measure called Predictive Diversity Score (PDS) which outperforms the existing bayesian Model Averaging (BMA). Experimental results across large-scale benchmarks (e.g., ImageNet) show the efficacy of the proposed framework.

**Strengths:**

- The research direction is promising, aiming to scale the current approach for large-scale data use while minimizing dependence on external OOD data.

- The analysis justifying the adaptive weights in Section 2.2 and quantifying diversity in Table 1 is well-executed.

- Additionally proposed components such as PDS, stochastic sum and shallow disagreement seem practical.

**Weaknesses:**

- In the proposed framework, each OOD sample is individually labeled during training. The reviewer’s main concern lies in conflicts within the ensemble, where simultaneous agreement (cross-entropy supervision) and disagreement arise over the same OOD data point. For instance, the same data can involve both agreement and disagreement terms during training.  Although the method beneficially induces OOD data from the training set, I would like to understand the authors' reasoning for this design choice, especially regarding the fluctuating membership status (ID or OOD) of the same data and why this approach may yield improvements over existing methods.

- According to the proposed framework, the membership status (i.e., ID or OOD) of each training data point may fluctuate throughout training, potentially even at each epoch. I believe analyzing this dynamic would enhance the manuscript, as it would be insightful to observe which data points are identified as hard samples (i.e., OOD data) by the trained models and how this impacts quantitative performance. For instance, an additional analysis could address how fluctuations in the amount of OOD data and the frequency of membership shifts for individual samples influence performance metrics at different stages. Furthermore, observing the characteristics of samples that consistently remain OOD or alternate between ID and OOD membership would provide valuable insights.

- The performance gains are modest compared to the baseline methods.

- The paper is a bit unkind; please clarify the meaning of "Div HPs" in the captions of the relevant tables, even though it is explained in the main text. Additionally, in Table 2, the amount by which the weight is raised is missing. Beyond listing the applied weight values, the reviewer suggests an ablation study to explore how varying weights impact quantitative performance, along with relevant discussions on the corresponding results.

**Questions:**

- As mentioned, the research direction is promising. However, obtaining unlabeled data, particularly for natural scene images (e.g., through web-crawling or platforms like YouTube), does not seem overly challenging. What distinguishes the proposed method from existing approaches that rely on explicit OOD data? For instance, the reviewer suggests including additional experiments in domains where obtaining suitable unlabeled OOD data is challenging, such as the medical field, to better demonstrate the advantages of the proposed framework over current methods.

- Is there a specific reason for omitting the results of “DivDis,” a fundamental baseline method, from Table 1?

- Why are the results from baseline methods, namely DivDis and A2D, missing in both the 5th and 6th rows of Table 2?

- The authors of DivDis have already demonstrated the viability of their method through parallel computation for diversification. What distinct advantages do stochastic sum and shallow disagreement offer over this approach?

- Please see the items in Weaknesses.

---

> ### Author Response · Authors · 2024-11-18
> **Words of gratitude + W1. Explain design choice: agreeing and disagreeing on the same datapoints.**
>
> We thank the reviewer for the high-quality review of our paper and for providing a detailed list of weaknesses and questions. Below we provide responses to them (some responses will appear a bit later than others due to the time needed for experiments running and terminology revision):
>
> ## W1. Explain design choice: agreeing and disagreeing on the same datapoints.
>
> Yes, for some samples, the two objectives clash. Here's our formulation for sample $n$ and model pair $(m,l)$: $L_{HDR} = L_{main} + \alpha_n L_{div}$. When $\alpha_n > 0$, both terms are applied to the same sample $n$, leading to potential clash in objectives. We control the relative importance of the terms through $\alpha_n$: for harder samples, we make $\alpha_n$ greater, such that the relative weight of the diversification term is greater and vice versa.
>
> What would have been cleaner is to control the weights like this: $L_{HDR} = (1-\alpha_n) L_{main} + \alpha_n L_{div}$, which is also possible indeed. We will run the corresponding experiments and post a separate response with them.

---

> ### Author Response · Authors · 2024-11-18
> **W2. Reasoning for design choice of fluctuating ID/OOD membership; Analyze dynamics of data points ID/OOD membership fluctuations during training.**
>
> The membership status of each sample often shifts during training, particularly in the initial epochs. This dynamic behaviour is beneficial. The role of hard samples, those with higher $\alpha_n$ values, is to improve the models' ability to generate diverse outputs. If these hard samples change throughout training, it is important to adapt continuously to maximise diversification.
>
> We will soon post a new response with the results of the progression of $\alpha_n$ values across datapoints to analyze ID/OOD membership fluctuations during training.

---

> > ### Author Response · Authors · 2024-11-18
> > **W3. Why does disagreement on hard samples yield improvements over existing methods?**
> >
> > Existing methods often assume that out-of-distribution (OOD) samples serve as diversity-inducing samples. While this assumption is frequently valid, it is not universally true. Even within datasets distinct from the training data, there are samples that remain close to the in-distribution training set. For such cases, models struggle to produce plausibly diverse outputs.
> >
> > Additionally, previous methods typically source diversity-inducing OOD samples from the same distribution as the OOD evaluation set, such as held-out waterbirds in [a, b]. This approach creates ambiguity. It becomes unclear whether the OOD dataset genuinely aids in diversifying ensembles or simply introduces information leakage from the evaluation set.
> >
> > Finally, reliance on OOD data raises practical concerns. It is often unclear which data should be selected for a given deployment scenario.
> >
> > To address these issues, we propose reusing the original training data. Our findings show that diversification does not require OOD data (e.g. IN-R doesn't help A2D and DivDis in Table 2). Instead, what is needed are hard samples where models can generate plausibly different predictions. Such samples are abundant in realistic training datasets like ImageNet.
> >
> >
> > [a] Yoonho Lee, Huaxiu Yao, and Chelsea Finn. Diversify and disambiguate: Out-of-distribution robustness via disagreement. In
> > International Conference on Learning Representations, 2023.
> >
> > [b] Matteo Pagliardini, Martin Jaggi, François Fleuret, and Sai Praneeth Karimireddy. Agree to disagree: Diversity through disagreement
> > for better transferability. In International Conference on Learning Representations, 2023.

---

> > > ### Author Response · Authors · 2024-11-18
> > > **W4. The performance gains are modest compared to the baseline methods.**
> > >
> > > We believe that providing the reference points for performance gains on the used benchmarks can help to remove the possible confusion. Papers that use ViT-B/16 architecture with ImageNet-scale usually provide 5-7 percent points (pp) increase while having around 40-50 percent of overall performance on IN-A and IN-R. For example, we can observe 7pp drop on IN-A and 0.3pp growth on IN-R (see Table 5 in [a]); 6.7pp growth on IN-A, 4.9pp growth on IN-R (see Table 5 in [b]); 6.3 growth on IN-R (see Table 2 in [c]).
> > >
> > > In our paper we have performance deltas of the same magnitude. Namely, if we compare deep ensemble with hyperparameter diversification (golden standard for ensemble diversification [d]) for ensemble with 50 models, then HDR provides 8pp boost from 45.8% to 53.8% and makes a significant difference.
> > > Similarly, 44.7% --> 48.7% accuracy on IN-R (from 1 model to 5 HDR models ensembles) is really a significant jump of 4pp.
> > >
> > > [a] Mao, X., Qi, G., Chen, Y., Li, X., Duan, R., Ye, S., He, Y., & Xue, H. (2021). Towards Robust Vision Transformer. 2022 IEEE/CVF Conference on Computer Vision and Pattern Recognition (CVPR), 12032-12041.
> > > [b] Herrmann, C., Sargent, K., Jiang, L., Zabih, R., Chang, H., Liu, C., Krishnan, D., & Sun, D. (2021). Pyramid Adversarial Training Improves ViT Performance. 2022 IEEE/CVF Conference on Computer Vision and Pattern Recognition (CVPR), 13409-13419.
> > > [c] Chen, X., Hsieh, C., & Gong, B. (2021). When Vision Transformers Outperform ResNets without Pretraining or Strong Data Augmentations. ArXiv, abs/2106.01548.
> > > [d] Wortsman, M., Ilharco, G., Gadre, S.Y., Roelofs, R., Gontijo-Lopes, R., Morcos, A.S., Namkoong, H., Farhadi, A., Carmon, Y., Kornblith, S., & Schmidt, L. (2022). Model soups: averaging weights of multiple fine-tuned models improves accuracy without increasing inference time. ArXiv, abs/2203.05482.

---

> > > > ### Author Response · Authors · 2024-11-18
> > > > **W5. Explain what "Div. HPs" is in captions.**
> > > >
> > > > We thank the reviewer for the suggestion and added the description to Tables 1, 2 and 3 in the updated pdf.

---

> ### Author Response · Authors · 2024-11-18
> **Q1. What distinguishes the proposed method from existing approaches that rely on explicit OOD data?**
>
> What matters for diversification is not sourcing "OOD samples" - it's the hard samples that help (e.g. IN-R doesn't help A2D and DivDis in Table 2).
>
> While OOD datasets might contain more informative hard samples than the training dataset, using them without proper filtering is not an option. Even within a different dataset than the training data, there exist samples that are close enough to in-distribution training data where models find it extremely difficult to propose plausibly diverse outputs (that is why baselines have low diversity in outputs even on OOD datasets, as can be seen in Table 1 - e.g. 1.04/5 unique predictions on average for ensemble of size 5 on IN-C-1).
>
> That fact was not so pronounced with the previous approaches utilizing external OOD data. The possible reason is that they source the diversity-inducing OOD samples from the same distribution as the OOD evaluation set (e.g. waterbirds held-out set in [a, b]). This leads to confounding of whether the OOD dataset is helping with the diversification of the ensembles or it is leaking potential eval-set information.
>
> Furthermore, even assuming that we decided to use OOD samples for disagreement, reliance on OOD data makes it unclear in practice which data should be chosen for each deployment scenario.
>
> We have revised the terminologies and precise argumentation in §2.2 of the updated PDF.
>
> [a] Yoonho Lee, Huaxiu Yao, and Chelsea Finn. Diversify and disambiguate: Out-of-distribution robustness via disagreement. In
> International Conference on Learning Representations, 2023.
>
> [b] Matteo Pagliardini, Martin Jaggi, François Fleuret, and Sai Praneeth Karimireddy. Agree to disagree: Diversity through disagreement
> for better transferability. In International Conference on Learning Representations, 2023.

---

> > ### Author Response · Authors · 2024-11-18
> > **Q2. Why omitting divdis from Table 1?**
> >
> > The main goal of the Table 1 is to show that HDR really diversifies the outputs. Since it is not a performance comparison we decided not to include all the baselines there for clarity. We kept only results for A2D as it enveloped the DivDis diversity results.
> >
> > Now we also provide the results for DivDis for completeness:
> >
> > | \hline Method | IN-Val             | IN-C-1             | IN-C-5             | iNat               | OI                 |
> > |---------------|--------------------|--------------------|--------------------|--------------------|--------------------|
> > | \hline DE     | 1.05               | 1.09               | 1.19               | 1.31               | 1.23               |
> > | +Div. HPs     | 1.04               | 1.11               | 1.32               | 1.48               | 1.33               |
> > | \hline A2D    | 1.11               | 1.04               | 1.15               | 1.19               | 1.91               |
> > | \hline DivDis    | 1.02               | 1.04               | 1.14               | 1.19               | 1.16               |
> > | \hline HDR    | $\mathbf{1 . 3 6}$ | $\mathbf{1 . 8 2}$ | $\mathbf{3 . 5 3}$ | $\mathbf{4 . 6 8}$ | $\mathbf{4 . 1 1}$ |

---

> ### Author Response · Authors · 2024-11-18
> **Q3 Why DivDis and A2D missing from rows in Table 2 + Q4 DivDis can be applied to parallel computations; What is the advantage of your method?**
>
> While the authors of divdis provided a code for parallel loss computation it can be seen that its complexity scales quadratically with the ensemble size (see line "marginal_p = torch.einsum("hd,ge->hgde", marginal_p, marginal_p)" in the code on page 16 [a]) as a consequence training an ensemble of big sizes on ImageNet is prohibitively expensive (for ensemble of size 50 one epoch takes more than 24 hours in our setup). Similar argument applies to A2D (quadratic complexity can be seen in line "for $ i \in  0, \ldots , m − 1$ do" of algorithm 2 in page 14 in [b]). For this exact reason, we did not train ensembles of such size for DivDis and A2D and did not include them in Table 2.
>
> Stochastic sum makes this loss scaling constant with respect to the ensemble size (diversification regularizer is computed for outputs of only two models for an ensemble of any size) and allows for training an ensemble of 50 models with 30 min per epoch. Shallow disagreement allows for further training speed up.
>
> [a] Yoonho Lee, Huaxiu Yao, and Chelsea Finn. Diversify and disambiguate: Out-of-distribution robustness via disagreement. In
> International Conference on Learning Representations, 2023.
>
> [b] Matteo Pagliardini, Martin Jaggi, François Fleuret, and Sai Praneeth Karimireddy. Agree to disagree: Diversity through disagreement for better transferability. In International Conference on Learning Representations, 2023.

---

> ### Author Response · Authors · 2024-11-21
> **W6. Add weights values to the table 2; sensitivity to lambda plot.**
>
> We thank the reviewer for the suggestion and will modify the revision with weights numbers and ablation study before 26.11

---

> > ### Author Response · Authors · 2024-11-26
> > **W1. Explain design choice: agreeing and disagreeing on the same datapoints vol. 2.**
> >
> > According to our quick verification, using another way to combine classification and diversification loss terms doesn't seem to make a significant difference in OOD generalization, while our default approach being better for OOD detection. We have included the result in the revision of §A.9.
> >
> > | Losses Combining | Val   | IN-A  | IN-R  | C-1   | C-5   | C-1   | C-5   | iNat  | OI    |
> > |-------------------|-------|-------|-------|-------|-------|-------|-------|-------|-------|
> > |  |  **Ensemble Acc.**     |   **Ensemble Acc.**    |  **Ensemble Acc.**     |  **Ensemble Acc.**     |    **AUROC**   | **AUROC** |   **AUROC**    |  **AUROC**     |   **AUROC**    |
> > | $L_{HDR} = (1-\alpha_n) L_{main} + \alpha_n L_{div}$        | **85.4** | 40.9  | 47.4  | **77.4** | **40.8** | 59.6  | 82.5  | 95.5  | 90.7  |
> > | $L_{HDR} =  L_{main} + \alpha_n L_{div}$           | 85.3  | **42.4** | **48.1** | 77.3  | 40.6  | **68.1** | **89.4** | **97.7** | **94.1** |

---

> > > ### Author Response · Authors · 2024-11-26
> > > **W2. Reasoning for design choice of fluctuating ID/OOD membership; Analyze dynamics of data points ID/OOD membership fluctuations during training vol. 2.**
> > >
> > > Following the reviewer's suggestion we have conducted an ID/OOD membership analysis for training samples. Here is our conclusion: as training progresses, the models gradually converge on a less-fluctuating (in comparison to easier samples) set of hard samples with higher $\alpha_n$ values. These values stabilize after a few epochs. For details on the progression of $\alpha_n$ values across datapoints, see Appendix §A.10 in the updated pdf.

---

> > > > ### Author Response · Authors · 2024-11-26
> > > > **W6. Add weights values to the table 2; sensitivity to lambda plot vol. 2.**
> > > >
> > > > We have added weights values to captions in Tables 2, 3 and 8 and provided results for the sensitivity study on how varying weights impact OOD generalization on DomainNet dataset in the plots for “DomainNet” paragraph of §A.5 in the updated pdf.

---

> ### Author Response · Authors · 2024-12-01
> **Thank you for your feedback!**
>
> Dear reviewer sLK4, thank you once again for your valuable feedback and for posing thoughtful, actionable questions. If any of our responses met your expectations, we would greatly appreciate it if you might consider revising your score. Should you have any further questions, please don’t hesitate to reach out—we’d be delighted to provide additional clarifications.

---

### Official Review · Reviewer_e4ou · 2024-11-05

**Soundness:** 2
**Presentation:** 4
**Contribution:** 2
**Rating:** 6
**Confidence:** 3

**Summary:**

This paper proposes a way to train an ensemble of models on OoD without access to Out-of-distribution data. Their proposed approach relies on determining hard samples from the training data and steering an ensemble of models to disagree on these samples. Authors claim this approach to be better for large-scale training. Randomness and last-layer training make their method efficient.

**Strengths:**

The paper is well-written and easy to follow. I especially like abstract. The method is quite efficient, especially as it requires training two models on each batch.

**Weaknesses:**

The novel contribution of the paper is not significant. It is not clear how the proposed hardness measure is useful for OoD. I have asked several questions below which will resolve my concerns.

**Questions:**

Authors argue:
>> With no OOD data, it is difficult to apply disagreement methods since the main training objective encourages all models to fit the training examples, hence to agree on all available data.

but then uses hard examples as a proxy for OoD data. Why would it make sense to replace OoD data with hard examples? Hard examples.


The authors use 2 models per batch for an ensemble of 50 models. This is quite interesting but requires more explanation as one would expect a number higher than 2 for effective diversification.


The main contribution of the paper is the introduction of a Hardness measure based on cross-entropy loss. However, there is no mention of previous works that have explored this direction (e.g., hard example mining is one direction that comes to mind). It would be interesting to mention previous works and explain how this work is different\new.

For OoD generalization, I am not sure how the proposed training makes a significant difference. Specifically, the results are quite close to a single model. But I am not sure what would be considered a significant improvement.


Another similar work that has explored Out-of-distribution generalization and an ensemble of models is the following: ZooD: Exploiting Model Zoo for Out-of-Distribution Generalization. Although this is in a different context.

Would it make sense to repeat some of the experiments on DomainNet (or a similar OoD dataset)?

---

> ### Author Response · Authors · 2024-11-18
> **Thank you + W1. Contribution is not significant: not clear how hardness measure helps OOD.**
>
> We thank the reviewer for the thorough review of our paper and provide our response to the named weaknesses and answers below (some responses will appear a bit later than others due to the time needed for experiments running and terminology revision):
>
> ## W1. Contribution is not significant: not clear how hardness measure helps OOD.
>
> Our goal is to increase ensemble diversity to improve OOD generalisation and detection (§3.3, §3.4). Achieving this requires data points that encourage models to disagree. Previous work has used a held-out OOD dataset for model disagreement, where the OOD dataset actually comes from the same distribution as the OOD evaluation set (§4.2 in [a], §4.1 in [b]). This is a major limitation of earlier approaches.
>
> We contribute our finding that for encouraging disagreement, what is important is not precisely the "OODness" but the "hardness" of the sample that invites diverse plausible hypotheses.
>
> From the practical point of view, using hard samples is more effective than relying on OOD samples. It is resource-efficient because all data points come from the original training set. Empirically, it produces more diverse ensembles and leads to better OOD generalisation and detection. For example, disagreeing on hard training samples via HDR instead of OOD set of IN-R via A2D leads to the growth from 1.19 to 4.68 in average number of unique predictions (shown in Table 1 to verify better diversity); growth from 45.2% to 48.7% in ensemble accuracy on IN-R (shown in Table 2 to verify better OOD generalization); growth from 85.0% to 89.4% in AUROC with PDS (shown in Table 3 to verify better OOD detection).
>
> [a] Yoonho Lee, Huaxiu Yao, and Chelsea Finn. Diversify and disambiguate: Out-of-distribution robustness via disagreement. In
> International Conference on Learning Representations, 2023.
>
> [b] Matteo Pagliardini, Martin Jaggi, François Fleuret, and Sai Praneeth Karimireddy. Agree to disagree: Diversity through disagreement
> for better transferability. In International Conference on Learning Representations, 2023.

---

> > ### Author Response · Authors · 2024-11-18
> > **Q1. Why would it make sense to replace OoD data with hard examples?**
> >
> > The reason OOD data are needed for diversification is not because they're out of distribution (i.e. out of the training dataset) but because some hard data points are needed to make the models diversify in plausible ways. Previous approaches sourced these data points from a separate dataset. We argue that such a dataset is not needed because eventually what we need are "hard" data points where models can plausibly differ in their responses. We have revised the terminologies and precise argumentation in §2.2 of the updated PDF.

---

> > > ### Author Response · Authors · 2024-11-18
> > > **Q4. Insignificant improvement in OOD generalization results.**
> > >
> > > We believe that providing the reference points for performance gains on the used benchmarks can help to remove the possible confusion. Papers that use ViT-B/16 architecture with ImageNet-scale usually provide 5-7 percent points (pp) increase while having around 40-50 percent of overall performance on IN-A and IN-R. For example, we can observe 7pp drop on IN-A and 0.3pp growth on IN-R (see Table 5 in [a]); 6.7pp growth on IN-A, 4.9pp growth on IN-R (see Table 5 in [b]); 6.3 growth on IN-R (see Table 2 in [c]).
> > >
> > > In our paper, we have performance deltas of the same magnitude. Namely, if we compare deep ensemble with hyperparameter diversification (golden standard for ensemble diversification [d]) for ensemble with 50 models, then HDR provides 8pp boost from 45.8% to 53.8% and makes a significant difference.
> > > Similarly, 44.7% --> 48.7% accuracy on IN-R (from 1 model to 5 HDR models ensembles) is really a significant jump of 4pp.
> > >
> > > [a] Mao, X., Qi, G., Chen, Y., Li, X., Duan, R., Ye, S., He, Y., & Xue, H. (2021). Towards Robust Vision Transformer. 2022 IEEE/CVF Conference on Computer Vision and Pattern Recognition (CVPR), 12032-12041.
> > > [b] Herrmann, C., Sargent, K., Jiang, L., Zabih, R., Chang, H., Liu, C., Krishnan, D., & Sun, D. (2021). Pyramid Adversarial Training Improves ViT Performance. 2022 IEEE/CVF Conference on Computer Vision and Pattern Recognition (CVPR), 13409-13419.
> > > [c] Chen, X., Hsieh, C., & Gong, B. (2021). When Vision Transformers Outperform ResNets without Pretraining or Strong Data Augmentations. ArXiv, abs/2106.01548.
> > > [d] Wortsman, M., Ilharco, G., Gadre, S.Y., Roelofs, R., Gontijo-Lopes, R., Morcos, A.S., Namkoong, H., Farhadi, A., Carmon, Y., Kornblith, S., & Schmidt, L. (2022). Model soups: averaging weights of multiple fine-tuned models improves accuracy without increasing inference time. ArXiv, abs/2203.05482.

---

> > > > ### Author Response · Authors · 2024-11-18
> > > > **Q5. Reference to similar work.**
> > > >
> > > > While the suggested paper indeed operates in a different context and focuses on model selection rather than ensemble diversification, we believe it is relevant to the application of ensembles to OOD generalisation. We included it in the "Ensemble" paragraph of §4 in the updated PDF.

---

> > > > > ### Author Response · Authors · 2024-11-21
> > > > > **Q3. Mention other hard samples mining methods.**
> > > > >
> > > > > Our approach to identifying hard samples in the training set is similar to hard sample mining methods used for worst-group robustness [a, b, c]. These methods aim to improve model performance on test samples from minority groups underrepresented in the training set (e.g. photos of animals in unusual contexts, such as a cow on a beach).
> > > > >
> > > > > While older worst-group robustness methods often required additional training labels for minority groups. The above mentioned approaches address this by upweighting the cross-entropy loss for the samples misclassified by a model preliminary trained with Empirical Risk Minimisation (ERM) on the full training dataset [a]. Extensions to this work include retraining only the last layer of the model [b] and using disagreements between multiple models in addition to misclassification to identify which samples to up-weight [c].
> > > > >
> > > > > Our approach differs. We do not require training a separate model with ERM first. We use hard samples for diversification, not for classification objectives.
> > > > >
> > > > > We have added this to the "Hard samples mining methods" paragraph of §4 in the updated PDF.
> > > > >
> > > > > [a] Liu, E.Z., Haghgoo, B., Chen, A.S., Raghunathan, A., Koh, P., Sagawa, S., Liang, P., & Finn, C. (2021). Just Train Twice: Improving Group Robustness without Training Group Information. ArXiv, abs/2107.09044.
> > > > >
> > > > > [b] Qiu, S., Potapczynski, A., Izmailov, P., & Wilson, A.G. (2023). Simple and Fast Group Robustness by Automatic Feature Reweighting. ArXiv, abs/2306.11074.
> > > > >
> > > > > [c] LaBonte, T., Muthukumar, V., & Kumar, A. (2023). Towards Last-layer Retraining for Group Robustness with Fewer Annotations. ArXiv, abs/2309.08534.

---

> ### Author Response · Authors · 2024-11-21
> **Q2. Why using 2 models per batch?**
>
> The usage of stochastic sum is justified by the fact that gradient of the loss in eq. 6 computed for only a subset of models is an unbiased estimator of the full gradient computed for all models. Therefore, gradient steps will move the ensemble weights in the optimal direction in expectation and training with stochastic sum is a valid approximation of the full ensemble training.
>
> In our experiments we used the smallest possible stochastic sum size of 2, as reducing it increases difference between ensemble members by diversifying the batches processed by each model during an epoch as well as reduces training time.
>
> Below we will explain why stochastic sum gradient is an unbiased estimator of the full gradient of eq. 6 and why reducing stochastic sum size leads to additional diversification as well as provide empirical results supporting our choice of the stochastic sum size.
>
> ### Stochastic sum is an unbiased estimator
>
> A simplified version of equation 6 for one batch of data is:
>
> $L = L_{main} + L_{div} = \frac{1}{|M|} \sum_{m \in M} L(m) + \frac{1}{|P_M|}\sum_{p \in P_{M}} G(p)$,
>
> where $M$ is a set of all models in ensemble; $L(m)$ is loss computed for the $m$-th model output; p is a pair of models; $P_M$ is a set of all possible pairs of models; $G(p)$ is a regularizer computed for the outputs of p.
>
> Let's imagine that on the current iteration we sampled a subset of models $I$. Then stochastic approximation of $\nabla L_{main}$ is:
>
> $\overline{\nabla L_{main}} = \frac{1}{|I|} \sum_{m \in I}\nabla L(m)$
>
>
> It is an unbiased estimate of the full gradient:
>
> $E_{m \in M}[\overline{\nabla L_{main}}] = \frac{1}{|I|} \sum_{m \in I} E_{m \in M}[\nabla L(m)] = \frac{1}{|I|} \cdot |I| \frac{1}{M}\sum_{m \in M} \nabla L(m) = \nabla [\frac{1}{M}\sum_{m \in M} L(m)] = \nabla L_{main}$
>
> Similarly, stochastic approximation of $\nabla L_{div}$ is:
>
> $\overline{\nabla L_{div}} = \frac{1}{|I|} \sum_{p \in I}\nabla G(p)$
>
> Which is also unbiased:
>
> $E_{m \in M}[\overline{\nabla L_{div}}] = \frac{1}{|I|} \sum_{p \in P_I}E_{m \in M}[\nabla G(p)] = \frac{1}{|I|} \cdot |I| \cdot \frac{1}{|P_M|} \sum_{p \in P_M}\nabla G(p) = \nabla L_{div}$,
>
> where $P_I$ is a set of all pairs of models from $I$.
>
> ### Reducing stochastic sum size leads to better diversity
>
> Smaller stochastic sums increase difference between ensemble members by diversifiying the subsets of the training data used to train each ensemble member. For example, with a stochastic sum of 2, each batch appears in the training subsets of only 2 models during one epoch. This is similar to bagging methods [a], where ensemble members benefit from training on different subsets of the training data.
>
> ### Empirical results support the choice of stochastic sum size 2
>
> Empirical results support the choice of stochastic sum size equal 2. As can be seen in Table 9 increasing stochastic sum size from 2 to 4 takes 8 times more time per epoch (453 vs 53 seconds) and results in worse OOD generalization performance across the board (e.g. IN-R accuracy drops from 48.1% to 46.8%) while giving only mixed gains on OOD detection (e.g. AUROC on IN-C-1 grows from 0.686 to 0.703 while AUROC on iNaturalist drops from 0.977 to 0.973).

---

> ### Author Response · Authors · 2024-11-21
> **Q6. Experiments on DomainNet.**
>
> Thank you for the suggestion we will conduct the experiments on DomainNet and add them to the rebuttal before 26.11

---

> > ### Author Response · Authors · 2024-11-26
> > **Q6. Experiments on DomainNet vol. 2.**
> >
> > We have conducted experiments on DomainNet and added their results to the paragraph “DomainNet” in §A.5 as well as in Table 8 in the updated PDF.
> >
> > Here is the copy of that table:
> >
> > |               Method        | Clip  | Info  | Paint | Real  | Quick | Sketch | Average |
> > |-----------------------------|-------|-------|-------|-------|-------|--------|---------|
> > | Deep Ensemble              |  58.2 |  18.4 |  47.7 | **58.4** |  14.9 |  50.9   |   41.4  |
> > | HDR             | **61.2** | **18.8** | **48.4** | **58.4** | **15.7** | **53.1** | **42.6** |
> >
> > HDR and Deep Ensemble have the same performance of $58.4\%$ on "Real" test set with HDR being better than Deep Ensemble on all other test sets (e.g. $61.2\%$ vs $58.2\%$ on "Clip" or $53.1\%$ vs $50.9\%$ on "Sketch") as well as in average performance ($42.6\%$ vs $41.4\%$).
> >
> > All ensemble training time was constrained to 1 hour per model (to fit in the rebuttal time budget); instead of the best checkpoint based on test accuracy we used the latest checkpoint for evaluation (to reflect the realistic scenario where test datasets are not available). That resulted in performances lower than state-of-the-art but sufficient to compare HDR and deep ensemble.

---

> ### Author Response · Authors · 2024-12-01
> **Thank you again for your feedback**
>
> Dear reviewer e4ou, thank you again for your feedback and for raising pertinent and actionable questions. If any of our response were satisfactory we would be extremely grateful if you could consider increasing your score. Please feel free to ask additional questions, we would be more than happy to provide further clarifications.

---

### Meta-Review · Area_Chair_MhED · 2024-12-22

**Metareview:**

This paper improves ensemble methods for out-of-distribution (OOD) detection and generalization by identifying hard samples within the training data instead of relying on external OOD data.

This paper received 3 reviews, with final ratings 5,6,5 from initial 5,5,5.  The authors provided extensive rebuttals, with additional experiments (e.g. on DomainNet) to address concerns about baseline comparisons, clarifications about the benefits of using hard samples over OOD data, emphasizing resource efficiency and scalability, updates to the manuscript including ablations and sensitivity analyses, and justifications for design choices, such as leveraging stochastic sums to improve training efficiency.  However, reviewers remain concerned about weak performance gains and conceptual clarity on the foundational assumption of using hard samples as OOD proxies.

While neither party fully clarified their positions, here is the crucial difference between the authors and the reviewers:  While an ensemble of diverse models could help with OOD generalization and detection, it is not necessarily the more diverse the ensemble is, the better the OOD performance; it needs to be the right kind of diversity.   The paper leveraged hard examples for achieving more diversity and equated more diversity with better OOD performance, but did not demonstrate convincingly -- through strong results or analysis -- that they indeed led to the right diversity for better OOD performance.   Therefore, the AC agrees with reviewers on their major concerns and generally lukewarm reaction to the paper and recommends rejection.

**Additional Comments On Reviewer Discussion:**

The authors' rebuttals were successful at improving one reviewer's rating from 5 to 6.   The other reviewers were pinged by the AC multiple times through the portal and personal emails, but unfortunately they did not respond.

---

### Decision · Program_Chairs · 2025-01-22

Reject